# Global Convergence and Variance Reduction for a Class of Nonconvex-Nonconcave Minimax Problems

**Junchi Yang**
UIUC
junchiy2@illinois.edu

**Negar Kiyavash**
EPFL
negar.kiyavash@epfl.ch

**Niao He**
UIUC & ETH Zurich
niao.he@inf.ethz.ch

## Abstract

Nonconvex minimax problems appear frequently in emerging machine learning applications, such as generative adversarial networks and adversarial learning. Simple algorithms such as the gradient descent ascent (GDA) are the common practice for solving these nonconvex games and receive lots of empirical success. Yet, it is known that these vanilla GDA algorithms with constant stepsize can potentially diverge even in the convex-concave setting. In this work, we show that for a subclass of nonconvex-nonconcave objectives satisfying a so-called two-sided Polyak-Łojasiewicz inequality, the alternating gradient descent ascent (AGDA) algorithm converges globally at a linear rate and the stochastic AGDA achieves a sublinear rate. We further develop a variance reduced algorithm that attains a provably faster rate than AGDA when the problem has the finite-sum structure.

## 1 Introduction

We consider minimax optimization problems of the forms

$$\min_{x \in \mathbb{R}^{d_1}} \max_{y \in \mathbb{R}^{d_2}} f(x, y) \tag{1}$$

where $f(x, y)$ is a possibly nonconvex-nonconcave function. Recent emerging applications in machine learning further stimulate a surge of interest in minimax problems. For example, generative adversarial networks (GANs) [23] can be viewed as a two-player game between a generator that produces synthetic data and a discriminator that differentiates between true and synthetic data. Other applications include reinforcement learning [9, 10, 11], robust optimization [42, 43], adversarial machine learning [54, 37], and so on. In many of these applications, $f(x, y)$ may be stochastic, namely, $f(x, y) = \mathbb{E}[F(x, y; \xi)]$, which corresponds to the expected loss of some random data $\xi$; or $f(x, y)$ may have the finite-sum structure, namely, $f(x, y) = \frac{1}{n} \sum_{i=1}^{n} f_i(x, y)$, which corresponds to the empirical loss over $n$ data points.

The most frequently used methods for solving minimax problems are the gradient descent ascent (GDA) algorithms (or their stochastic variants), with either simultaneous or alternating updates of the primal-dual variables, referred to as SGDA and AGDA, respectively. While these algorithms have received much empirical success especially in adversarial training, it is known that GDA algorithms with constant stepsizes could fail to converge even for the bilinear games [22, 40]; when they do converge, the stable limit point may not be a local Nash equilibrium [13, 38]. On the other hand, GDA algorithms can converge linearly to the saddle point for strongly-convex-strongly-concave functions [17]. Moreover, for many simple nonconvex-nonconcave objective functions, such as, $f(x, y) = x^2 + 3 \sin^2 x \sin^2 y - 4y^2 - 10 \sin^2 y$, we observe that GDA algorithms with constant

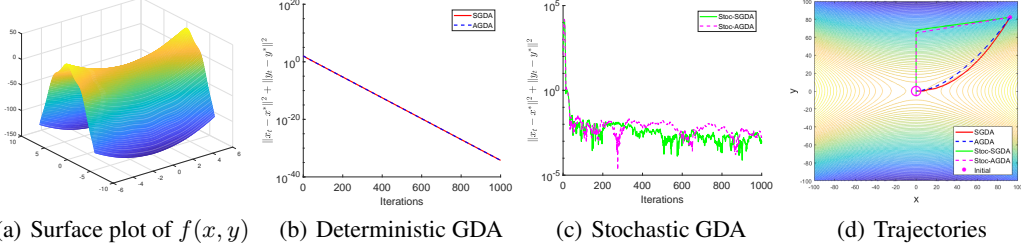

(a) Surface plot of $f(x, y)$  (b) Deterministic GDA  (c) Stochastic GDA  (d) Trajectories

Figure 1: (a) Surface plot of the nonconvex-nonconcave function $f(x, y) = x^2 + 3\sin^2 x \sin^2 y - 4y^2 - 10\sin^2 y$ ; (b) Convergence of SGDA and AGDA; (c) Convergence of stochastic SGDA and stochastic AGDA; (d) Trajectories of four algorithms

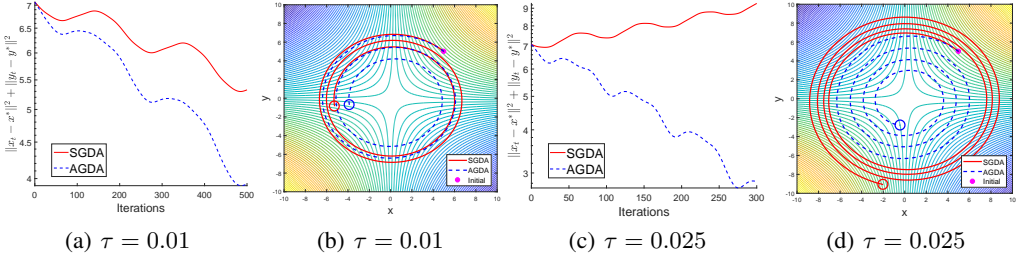

(a) $\tau = 0.01$  (b) $\tau = 0.01$  (c) $\tau = 0.025$  (d) $\tau = 0.025$

Figure 2: Consider $f(x, y) = \log(1 + e^x) + 3xy - \log(1 + e^y)$: (a) Convergence of AGDA and SGDA with the stepsize $\tau = 0.01$; (b) Trajectories of two algorithms with $\tau = 0.01$; (c) Convergence of AGDA and SGDA with stepsize $\tau = 0.025$; (d) Trajectories of two algorithms with $\tau = 0.025$;

stepsizes converge to the global Nash equilibrium (see Figure 1). These facts naturally raise a question: *Is there a general condition under which GDA algorithms converge to the global optima?*

Furthermore, the use of variance reduction techniques has played a prominent role in improving the convergence over stochastic or batch algorithms for both convex and nonconvex minimization problems [27, 52, 53, 58]. However, when it comes to the minimax problems, there are limited results, except under convex-concave setting [49, 15]. This leads to another open question: *Can we improve GDA algorithms for nonconvex-nonconcave minimax problems?*

## 1.1 Our contributions

In this paper, we address these two questions and specifically focus on the alternating gradient descent ascent, namely AGDA. This is due to several considerations. First of all, using alternating updates of GDA is more stable than simultaneous updates [22, 2] and often converges faster in practice. Note that for a convex-concave matrix game, SGDA may diverge while AGDA is proven to always have bounded iterates [22]. See Figure 2 for a simple illustration. Secondly, AGDA is widely used for training GANs and other minimax problems in practice; see e.g., [33, 41]. Yet there is a lack of discussion on the convergence of AGDA for general minimax problems in the literature, even for the favorable strongly-convex-strongly-concave setting. Alternating updating schemes are perceived more challenging to analyze than simultaneous updates; the latter treats two variables equally and has been extensively studied in vast literature of variational inequality. Our main contributions are summarized as follows.

**Two-sided PL condition.** First, we identity a general condition that relaxes the convex-concavity requirement of the objective function while still guaranteeing global convergence of AGDA and stochastic AGDA (Stoc-AGDA). We call this the two-sided PL condition, which requires that both players' utility functions satisfy Polyak-Łojasiewicz (PL) inequality [50]. The two-sided PL condition is very general and is satisfied by many important classes of functions: (a) all strongly-convex-strongly-concave functions; (b) all PL-strongly-concave function (discussed in [24]) and (c) many nonconvex-nonconcave objectives. Such conditions also hold true for various applications, including robust least square, generative adversarial imitation learning for linear quadratic regulator (LQR) dynamics [5], zero-sum linear quadratic game [63], and potentially many others in adversarial learning [14], robust phase retrieval [55, 64], robust control [18], and etc. We first investigate the landscape of objectives under the two-sided PL condition. In particular, we show that three notions of optimality: saddle point, minimax point, and stationary point are equivalent.

**Global convergence of AGDA.** We show that under the two-sided PL condition, AGDA with proper constant stepsizes converges globally to a saddle point at a linear rate of $\mathcal{O}(1 - \kappa^{-3})^t$, while Stoc-AGDA with proper diminishing stepsizes converges to a saddle point at a sublinear rate of $\mathcal{O}(\kappa^5/t)$, where $\kappa$ is the underlying condition number. To the best of our knowledge, this is the first result on the global convergence of a class of nonconvex-nonconvex problems. In contrast, most previous work deals with nonconvex-concave problems and obtains convergence to stationary points. On the other hand, because all strongly-convex-strongly-concave and PL-strongly-concave functions naturally satisfy the two-sided PL condition, our analysis fills the theoretical gap with the first convergence results of AGDA under these settings.

**Variance reduced algorithm.** For minimax problems with the finite-sum structure, we introduce a variance-reduced AGDA algorithm (VR-AGDA) that leverages the idea of stochastic variance reduced gradient (SVRG) [27, 52] with the alternating updates. We prove that VR-AGDA achieves the complexity of $\mathcal{O}\left((n + n^{2/3}\kappa^3)\log(1/\epsilon)\right)$, which improves over the $\mathcal{O}\left(n\kappa^3\log\frac{1}{\epsilon}\right)$ complexity of AGDA and the $\mathcal{O}\left(\kappa^5/\epsilon\right)$ complexity of Stoc-AGDA when applied to finite-sum minimax problems. Our numerical experiments further demonstrate that VR-AGDA performs significantly better than AGDA and Stoc-AGDA, especially for problems with large condition numbers. To our best knowledge, this is the first work to provide a variance-reduced algorithm and theoretical guarantees in the nonconvex-nonconcave regime of minimax optimization. In contrast, most previous variance-reduced algorithms require full or partial strong convexity and only apply to simultaneous updates.

**Nonconvex-PL games.** Lastly, as a side contribution, we show that for a broader class of nonconvex-nonconcave problems under only one-sided PL condition, AGDA converges to a $\epsilon$-stationary point within $\mathcal{O}(\epsilon^{-2})$ iterations, thus is optimal among all first-order algorithms. Our result shaves off a logarithmic factor of the best-known rate achieved by the multi-step GDA algorithm [47]. This directly implies the same convergence rate on nonconvex-strongly-concave objectives, and to our best knowledge, we are the first to show the convergence of AGDA on this class of functions. Due to page limitation, we defer this result to Appendix **??**.

## 1.2 Related work

**Nonconvex minimax problems.** There has been a recent surge in research on solving minimax optimization beyond the convex-concave regime [54, 8, 51, 56, 30, 47, 1, 32, 3, 48], but they differ from our work from various perspectives. Most of these work focus on the nonconvex-concave regime and aim for convergence to stationary points of minimax problems [8, 54, 31, 56]. Algorithms in these work require solving the inner maximization or some sub-problems with high accuracy, which are different from AGDA. Lin et al. [30] proposed an inexact proximal point method to find an $\epsilon$-stationary point for a class of weakly-convex-weakly-concave minimax problems. Their convergence result relies on assuming the existence of a solution to the corresponding Minty variational inequality, which is hard to verify. Abernethy et al. [1] showed the linear convergence of a second-order iterative algorithm, called Hamiltonian gradient descent, for a subclass of "sufficiently bilinear" functions. Very recently, Xu et al. [60] and Boţ and Böhm [4] anslyze AGDA in nonconvex-(strongly-)concave setting. There is also a line of work in understanding the dynamics in minimax games [39, 20, 19, 21, 12, 25].

**Variance-reduced minimax optimization.** Palaniappan and Bach [49], Luo et al. [34], Chavdarova et al. [7] provided linear-convergent algorithms for strongly-convex-strongly-concave objectives, based on simultaneous updates. Du and Hu [15] extended the result to convex-strongly-concave objectives with full-rank coupling bilinear term. In contrast, we are dealing with a much broader class of objectives that are possibly nonconvex-nonconcave. We point out that Luo et al. [35] and Xu et al. [59] recently introduced variance-reduced algorithms for finding the stationary point of nonconvex-strongly-concave problems, which is again different from our setting.

## 2 Global optima and two-sided PL condition

Throughout this paper, we assume that the function $f(x, y)$ in (1) is continuously differentiable and has Lipschitz gradient. Here $\|\cdot\|$ is used to denote the Euclidean norm.

**Assumption 1** (Lipschitz gradient). *There exists a positive constant $l > 0$ such that*

$$\max\{\|\nabla_y f(x_1, y_1) - \nabla_y f(x_2, y_2)\|, \|\nabla_x f(x_1, y_1) - \nabla_x f(x_2, y_2)\|\} \leq l[\|x_1 - x_2\| + \|y_1 - y_2\|],$$

*holds for all $x_1, x_2 \in \mathbb{R}^{d_1}, y_1, y_2 \in \mathbb{R}^{d_2}$.*

We now define three notions of optimality for minimax problems. The most direct notion of optimality is global minimax point, at which $x^*$ is an optimal solution to the function $g(x) := \max_y f(x, y)$ and $y^*$ is an optimal solution to $\max_y f(x^*, y)$. In the two-player zero-sum game, the notion of saddle point is also widely used [57, 44]. For a saddle point $(x^*, y^*)$, $x^*$ is an optimal solution to $\min_x f(x, y^*)$ and $y^*$ is an optimal solution to $\max_y f(x^*, y)$.

**Definition 1** (Global optima)**.**

1. *$(x^*, y^*)$ is a global minimax point, if for any $(x, y): f(x^*, y) \leq f(x^*, y^*) \leq \max_{y'} f(x, y')$.*

2. *$(x^*, y^*)$ is a saddle point, if for any $(x, y): f(x^*, y) \leq f(x^*, y^*) \leq f(x, y^*)$.*

3. *$(x^*, y^*)$ is a stationary point, if: $\nabla_x f(x^*, y^*) = \nabla_y f(x^*, y^*) = 0$.*

For general nonconvex-nonconcave minimax problems, these three notions of optimality are not necessarily equivalent. A stationary point may not be a saddle point or a global minimax point; a global minimax point may not be a saddle point or a stationary point. Note that for minimax problems, a saddle point or a global minimax point may not always exist. However, since our goal in this paper is to find global optima, in the remainder of the paper, we assume that a saddle point always exists.

**Assumption 2** (Existence of saddle point)**.** *The objective function $f$ has at least one saddle point. We also assume that for any fixed $y$, $\min_{x \in \mathbb{R}^{d_1}} f(x, y)$ has a nonempty solution set and a optimal value, and for any fixed $x$, $\max_{y \in \mathbb{R}^{d_2}} f(x, y)$ has a nonempty solution set and a finite optimal value.*

For unconstrained minimization problems: $\min_{x \in \mathbb{R}^n} f(x)$, Polyak [50] proposed Polyak-Łojasiewicz (PL) condition, which is sufficient to show global linear convergence for gradient descent without assuming convexity. Specifically, a function $f(\cdot)$ satisfies PL condition if it has a nonempty solution set and a finite optimal value $f^*$, and there exists some $\mu > 0$ such that $\|\nabla f(x)\|^2 \geq 2\mu(f(x) - f^*), \forall x$. As discussed in Karimi et al. [28], PL condition is weaker, or not stronger, than other well-known conditions that guarantee linear convergence for gradient descent, such as error bounds (EB) [36], weak strong convexity (WSC) [45] and restricted secant inequality (RSI) [61].

We introduce a straightforward generalization of the PL condition to the minimax problem: function $f(x, y)$ satisfies the PL condition with constant $\mu_1$ with respect to $x$, and $-f$ satisfies PL condition with constant $\mu_2$ with respect to $y$. We formally state this in the following definition.

**Definition 2** (Two-sided PL condition)**.** *A continuously differentiable function $f(x, y)$ satisfies the two-sided PL condition if there exist constants $\mu_1, \mu_2 > 0$ such that: $\forall x, y$,*

$$\|\nabla_x f(x, y)\|^2 \geq 2\mu_1 [f(x, y) - \min_x f(x, y)], \quad \|\nabla_y f(x, y)\|^2 \geq 2\mu_2 [\max_y f(x, y) - f(x, y)].$$

The two-sided PL condition does not imply convexity-concavity, and it is a much weaker condition than strong-convexity-strong-concavity. In Lemma 2.1, we show that three notions of optimality are equivalent under the two-sided PL condition. Note that they may not be unique.

**Lemma 2.1.** *If the objective function $f(x, y)$ satisfies the two-sided PL condition, then the following holds true:*

*(saddle point) $\Leftrightarrow$ (global minimax) $\Leftrightarrow$ (stationary point).*

Below we give some examples that satisfy this condition.

**Example 1.** *The nonconvex-nonconcave function in the introduction, $f(x, y) = x^2 + 3\sin^2 x \sin^2 y - 4y^2 - 10\sin^2 y$ satisfies the two-sided PL condition with $\mu_1 = 1/16, \mu_2 = 1/11$ (see Appendix **??**).*

**Example 2.** *$f(x, y) = F(Ax, By)$, where $F(\cdot, \cdot)$ is strongly-convex-strongly-concave and $A$ and $B$ are arbitrary matrices, satisfies the two-sided PL condition.*

**Example 3.** *The generative adversarial imitation learning for LQR can be formulated as $\min_K \max_\theta m(K, \theta)$, where $m$ is strongly-concave in terms of $\theta$ and satisfies PL condition in terms of $K$ (see [5] for more details), thus satisfying the two-sided PL condition.*

**Example 4.** *In a zero-sum linear quadratic (LQ) game, the system dynamics are characterized by $x_{t+1} = Ax_t + Bu_t + Cv_t$, where $x_t$ is the system state and $u_t, v_t$ are the control inputs from two-players. After parameterizing the policies of two players by $u_t = -Kx_t$ and $v_t = -Lx_t$, the*

*value function is $C(K, L) = \mathbb{E}_{x_0 \sim \mathcal{D}} \left\{ \sum_{t=0}^{\infty} \left[ x_t^\top Q x_t + (K x_t)^\top R^u (K x_t) - (L x_t)^\top R^v (L x_t) \right] \right\}$, where $\mathcal{D}$ is the distribution of the initial state $x_0$ (see [63] for more details). Player 1 (player 2) wants to minimize (maximize) $C(K, L)$, and the game is formulated as $\min_K \max_L C(K, L)$. Fixing $L$ (or $K$), $C(\cdot, L)$ (or $-C(K, \cdot)$) becomes an objective of an LQR problem, and therefore satisfies the PL condition [18] when $\text{argmin}_K C(K, L)$ and $\text{argmax}_L C(K, L)$ are well-defined.*

The two-sided PL condition includes rich classes of functions, including: (a) all strongly-convex-strongly-concave functions; (b) some convex-concave functions (e.g., Example 2); (c) some nonconvex-strongly-concave functions (e.g., Example 3); (d) some nonconvex-nonconcave functions (e.g., Example 1 and 4). Under the two-sided PL condition, the function $g(x) := \max_y f(x, y)$ satisfies PL condition with $\mu_1$ (see Appendix **??**). Moreover, it holds that $g$ is also $L$-smooth with $L := l + l^2/\mu_2$ [47]. Finally, we denote $\mu = \min(\mu_1, \mu_2)$ and $\kappa = \frac{l}{\mu}$, which represents the condition number of the problem.

## 3 Global convergence of AGDA and Stoc-AGDA

In this section, we establish the convergence rate of the stochastic alternating gradient descent ascent (Stoc-AGDA) algorithm, which we present in Algorithm 1, under the two-sided PL condition. Stoc-AGDA updates variables $x$ and $y$ sequentially using stochastic gradient descent/ascent steps. Here we make standard assumptions about stochastic gradients $G_x(x, y, \xi)$ and $G_y(x, y, \xi)$.

**Assumption 3** (Bounded variance). *$G_x(x, y, \xi)$ and $G_y(x, y, \xi)$ are unbiased stochastic estimators of $\nabla_x f(x, y)$ and $\nabla_y f(x, y)$ and have variances bounded by $\sigma^2 > 0$.*

---

**Algorithm 1** Stoc-AGDA

---

1: Input: $(x_0, y_0)$, stepsizes $\{\tau_1^t\}_t > 0, \{\tau_2^t\}_t > 0$
2: **for all** $t = 0, 1, 2, ...$ **do**
3:     Draw two i.i.d. samples $\xi_{t1}, \xi_{t2} \sim P(\xi)$
4:     $x_{t+1} \leftarrow x_t - \tau_1^t G_x(x_t, y_t, \xi_{t1})$
5:     $y_{t+1} \leftarrow y_t + \tau_2^t G_y(x_{t+1}, y_t, \xi_{t2})$
6: **end for**

---

Note that Stoc-AGDA with constant stepsizes (i.e., $\tau_1^t = \tau_1$ and $\tau_2^t = \tau_2$) and noiseless stochastic gradient (i.e., $\sigma^2 = 0$) reduces to AGDA:

$$
\begin{aligned}
x_{t+1} &= x_t - \tau_1 \nabla_x f(x_t, y_t), \\
y_{t+1} &= y_t + \tau_2 \nabla_y f(x_{t+1}, y_t).
\end{aligned}
\tag{2}
$$

We measure the inaccuracy of $(x_t, y_t)$ through the potential function

$$
P_t := a_t + \lambda \cdot b_t,
\tag{3}
$$

where $a_t = \mathbb{E}[g(x_t) - g^*], b_t = \mathbb{E}[g(x_t) - f(x_t, y_t)]$ and the balance parameter $\lambda > 0$ will be specified later in the theorems. Recall that $g(x) := \max_y f(x, y)$ and $g^* := \min_x g(x)$. This metric is driven by the definition of minimax point, because $g(x) - g^*$ and $g(x) - f(x, y)$ are non-negative for any $(x, y)$, and both equal to 0 if and only if $(x, y)$ is a minimax point.

**Stoc-AGDA with constant stepsizes** We first consider Stoc-AGDA with constant stepsizes. We show that $\{(x_t, y_t)\}_t$ will converge linearly to a neighbourhood of the optimal set.

**Theorem 3.1.** *Suppose Assumptions 1, 2, 3 hold and $f(x, y)$ satisfies the two-sided PL condition with $\mu_1$ and $\mu_2$. Define $P_t := a_t + \frac{1}{10} b_t$. If we run Algorithm 1 with $\tau_2^t = \tau_2 \leq \frac{1}{l}$ and $\tau_1^t = \tau_1 \leq \frac{\mu_2^2 \tau_2}{18l^2}$,*

$$
P_t \leq (1 - \frac{1}{2} \mu_1 \tau_1)^t P_0 + \delta,
\tag{4}
$$

*where $\delta = \frac{(1 - \mu_2 \tau_2)(L+l)\tau_1^2 + l\tau_2^2 + 10L\tau_1^2}{10\mu_1 \tau_1} \sigma^2$.*

**Remark 1.** *In the theorem above, we choose $\tau_1$ smaller than $\tau_2$, $\tau_1/\tau_2 \leq \mu_2^2/(18l^2)$, because our potential function is not symmetric about $x$ and $y$. Another reason is because we want $y_t$*

to approach $y^*(x_t) \in \arg\max_y f(x_t, y)$ faster so that $\nabla_x f(x_t, y_t)$ is a better approximation for $\nabla g(x_t)$ ($\nabla g(x) = \nabla_x f(x, y^*(x))$), see Nouiehed et al. [47]). Indeed, it is common to use different learning rates for $x$ and $y$ in GDA algorithms for nonconvex minimax problems; see e.g., Jin et al. [26] and Lin et al. [31]. Note that the ratio between these two learning rates is quite crucial here. We also observe empirically when the same learning rate is used, even if small, the algorithm may not converge to saddle points.

**Remark 2.** When $t \to \infty$, $P_t \to \delta$. If $\tau_1 \to 0$ and $\tau_2^2/\tau_1 \to 0$, the error term $\delta$ will go to 0. When using smaller stepsizes, the algorithm reaches a smaller neighbour of the saddle point yet at the cost of a slower rate, as the contraction factor also deteriorates.

**Linear convergence of AGDA**    Setting $\sigma^2 = 0$, it follows immediately from the previous theorem that AGDA converges linearly under the two-sided PL condition. Moreover, we have the following:

**Theorem 3.2.** Suppose Assumptions 1, 2 hold and $f(x,y)$ satisfies the two-sided PL condition with $\mu_1$ and $\mu_2$. Define $P_t := a_t + \frac{1}{10}b_t$. If we run AGDA with $\tau_1 = \frac{\mu_2^2}{18l^3}$ and $\tau_2 = \frac{1}{l}$, then

$$P_t \leq \left(1 - \frac{\mu_1\mu_2^2}{36l^3}\right)^t P_0. \tag{5}$$

Furthermore, $\{(x_t, y_t)\}_t$ converges to some saddle point $(x^*, y^*)$, and

$$\|x_t - x^*\|^2 + \|y_t - y^*\|^2 \leq \alpha \left(1 - \frac{\mu_1\mu_2^2}{36l^3}\right)^t P_0, \tag{6}$$

where $\alpha$ is a constant depending on $\mu_1, \mu_2$ and $l$.

The above theorem implies that the limit point of $\{(x_t, y_t)\}_t$ is a saddle point and the distance to the saddle point decreases in the order of $\mathcal{O}\left((1 - \kappa^{-3})^t\right)$. Note that in the special case when the objective is strongly-convex-strongly-concave, it is known that SGDA (GDA with simultaneous updates) achieves an $\mathcal{O}(\kappa^2 \log(1/\epsilon))$ iteration complexity (see, e.g., Facchinei and Pang [17]) and this can be further improved to match the lower complexity bound $\mathcal{O}(\kappa \log(1/\epsilon))$ [62] by extragradient methods [29] or Nesterov's dual extrapolation [46]. However, these results heavily rely on the strong monotonicity of the corresponding variational inequality, which does not apply here. Our analysis technique is totally different. Since the general two-sided PL condition contains a much broader class of functions, we do not expect to achieve the same dependency on $\kappa$, especially for a simple algorithm like AGDA. Note that even the multi-step GDA in [47] results in the same $\kappa^3$ dependency, but without linear convergence rate. Hence, our conjecture is that the $\kappa^3$ dependency of AGDA can not be improved without modifying the algorithm. We leave this investigation for future work.

**Stoc-AGDA with diminishing stepsizes**    While Stoc-AGDA with constant stepsizes only converges linearly to a neighbourhood of the saddle point, Stoc-AGDA with diminishing stepsizes converges to the saddle point but at a sublinear rate $\mathcal{O}(1/t)$.

**Theorem 3.3.** Suppose Assumptions 1, 2, 3 hold and $f(x,y)$ satisfies the two-sided PL condition with $\mu_1$ and $\mu_2$. Define $P_t = a_t + \frac{1}{10}b_t$. If we run algorithm 1 with stepsizes $\tau_1^t = \frac{\beta}{\gamma+t}$ and $\tau_2^t = \frac{18l^2\beta}{\mu_2^2(\gamma+t)}$ for some $\beta > 2/\mu_1$ and $\gamma > 0$ such that $\tau_1^1 \leq \min\{1/L, \mu_2^2/18l^2\}$, then we have

$$P_t \leq \frac{\nu}{\gamma+t}, \quad \text{where } \nu := \max\left\{\gamma P_0, \frac{[(L+l)\beta^2 + 18^2l^5\beta^2/\mu_2^4 + 10L\beta^2]\sigma^2}{10\mu_1\beta - 20}\right\}. \tag{7}$$

**Remark 3.** Note the rate is affected by $\nu$, and the first term in the definition of $\nu$ is controlled by the initial point. In practice, we can find a good initial point by running Stoc-AGDA with constant stepsizes so that only the second term in the definition of $\nu$ matters. Then by choosing $\beta = 3/\mu_1$, we have $\nu = \mathcal{O}\left(\frac{l^5\sigma^2}{\mu_1^2\mu_2^4}\right)$. Thus, the convergence rate of Stoc-AGDA is $\mathcal{O}\left(\frac{\kappa^5\sigma^2}{\mu t}\right)$.

## 4   Stochastic variance-reduced AGDA algorithm

In this section, we study the minimax problem with the finite-sum structure: $\min_x \max_y f(x,y) = \frac{1}{n}\sum_{i=1}^n f_i(x,y)$, which arises ubiquitously in machine learning. We are especially interested in the

---

**Algorithm 2** VR-AGDA

1: input: $(\tilde{x}_0, \tilde{y}_0)$, stepsizes $\tau_1, \tau_2$, iteration numbers $N, T$
2: **for all** $k = 0, 1, 2, ...$ **do**
3:     **for all** $t = 0, 1, 2, ...T - 1$ **do**
4:         $x_{t,0} = \tilde{x}_t, \quad y_{t,0} = \tilde{y}_t,$
5:         compute $\nabla_x f(\tilde{x}_t, \tilde{y}_t) = \frac{1}{n} \sum_{i=1}^{n} \nabla_x f_i(\tilde{x}_t, \tilde{y}_t)$ and $\nabla_y f(\tilde{x}_t, \tilde{y}_t) = \frac{1}{n} \sum_{i=1}^{n} \nabla_y f_i(\tilde{x}_t, \tilde{y}_t)$
6:         **for all** $j = 0$ to $N - 1$ **do**
7:             sample i.i.d. indices $i_j^1, i_j^2$ uniformly from $[n]$
8:             $x_{t,j+1} = x_{t,j} - \tau_1[\nabla_x f_{i_j^1}(x_{t,j}, y_{t,j}) - \nabla_x f_{i_j^1}(\tilde{x}_t, \tilde{y}_t) + \nabla_x f(\tilde{x}_t, \tilde{y}_t)]$
9:             $y_{t,j+1} = y_{t,j} + \tau_2[\nabla_y f_{i_j^2}(x_{t,j+1}, y_{t,j}) - \nabla_y f_{i_j^2}(\tilde{x}_t, \tilde{y}_t) + \nabla_y f(\tilde{x}_t, \tilde{y}_t)]$
10:         **end for**
11:         $\tilde{x}_{t+1} = x_{t,N}, \quad \tilde{y}_{t+1} = y_{t,N}$
12:     **end for**
13:     choose $(x^k, y^k)$ from $\{\{(x_{t,j}, y_{t,j})\}_{j=0}^{N-1}\}_{t=0}^{T-1}$ uniformly at random
14:     $\tilde{x}_0 = x^k, \quad \tilde{y}_0 = y^k$
15: **end for**

---

case when $n$ is large. We assume the overall objective function $f(x, y)$ satisfies the two-sided PL condition with $\mu_1$ and $\mu_2$, but do not assume each $f_i$ to satisfy the two-sided PL condition. Instead of Assumption 1, we assume each component $f_i$ has Lipschitz gradients.

**Assumption 4.** *Each $f_i$ has l-Lipschitz gradients.*

If we run AGDA with full gradients to solve the finite-sum minimax problem, the total complexity for finding an $\epsilon$-optimal solution is $\mathcal{O}(n\kappa^3 \log(1/\epsilon))$ by Theorem 3.2. Despite the linear convergence, the per-iteration cost is high and the complexity can be huge when the number of components $n$ and condition number $\kappa$ are large. Instead, if we run Stoc-AGDA, this leads to the total complexity $\mathcal{O}\left(\frac{\kappa^5 \sigma^2}{\mu_2 \epsilon}\right)$ by Remark 3, which has worse dependence on $\epsilon$.

Motivated by the recent success of stochastic variance reduced gradient (SVRG) technique [27, 52, 49], we introduce the VR-AGDA algorithm (presented in Algorithm 2), that combines AGDA with SVRG so that the linear convergence is preserved while improving the dependency on $n$ and $\kappa$. VR-AGDA can be viewed as the applying SVRG to AGDA with restarting: at every epoch $k$, we restart the SVRG subroutine by initializing it with $(x^k, y^k)$, which is randomly selected from previous SVRG subroutine. This is partly inspired by the GD-SVRG algorithm for minimizing PL functions [52]. Notice when $T = 1$, VR-AGDA reduces to a double-loop algorithm which is similar to the SVRG for saddle point problems proposed by Palaniappan and Bach [49], except for several notable differences: (i) we are using the alternating updates rather than simultaneous updates, (ii) as a result, we require to sample two independent indices rather than one at each iteration, and (iii) most importantly, we are dealing with possibly nonconvex-nonconcave objectives that satisfy the two-sided PL condition. The following two theorems capture the convergence of VR-AGDA under different parameter setups.

**Theorem 4.1.** *Suppose Assumptions 2 and 4 hold and $f(x, y)$ satisfies the two-sided PL condition with $\mu_1$ and $\mu_2$. Define $P_k = a^k + \frac{1}{20}b^k$, where $a^k = \mathbb{E}[g(x^k) - g^*]$ and $b^k = \mathbb{E}[g(x^k) - f(x^k, y^k)]$. If we run VR-AGDA with $\tau_1 = \beta/(28\kappa^8 l)$, $\tau_2 = \beta/(l\kappa^6)$, $N = \lfloor \alpha\beta^{-2/3}\kappa^9(2 + 4\beta^{1/2}\kappa^{-3})^{-1}\rfloor$ and $T = 1$, where $\alpha, \beta$ are constants irrelevant to $l, n, \mu_1, \mu_2$, then $P_{k+1} \le \frac{1}{2}P_k$. This implies complexity of*

$$\mathcal{O}\big((n + \kappa^9) \log(1/\epsilon)\big)$$

*total for VR-AGDA to achieve an $\epsilon$-optimal solution.*

**Theorem 4.2.** *Under the same assumptions as Theorem 4.1 , if we run VR-AGDA with $\tau_1 = \beta/(28\kappa^2 l n^{2/3})$, $\tau_2 = \beta/(ln^{2/3})$, $N = \lfloor \alpha\beta^{-2/3}n(2 + 4\beta^{1/2}n^{-1/3})^{-1}\rfloor$, and $T = \lceil \kappa^3 n^{-1/3}\rceil$, where $\alpha, \beta$ are constants irrelevant to $l, n, \mu_1, \mu_2$, then $P_{k+1} \le \frac{1}{2}P_k$. This implies complexity of*

$$\mathcal{O}\big((n + n^{2/3}\kappa^3) \log(1/\epsilon)\big)$$

*for VR-AGDA to achieve an $\epsilon$-optimal solution.*

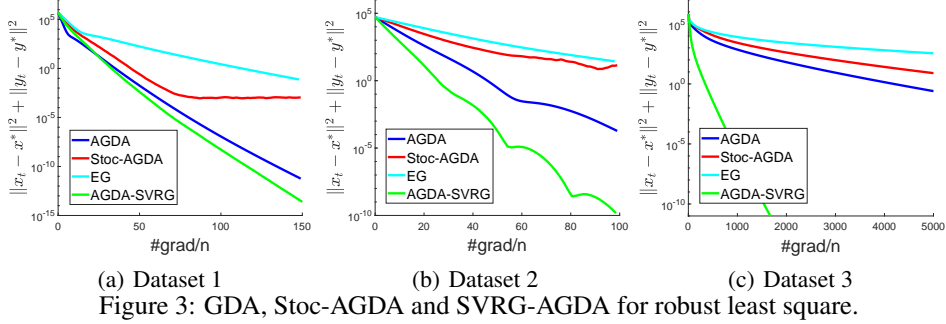

(a) Dataset 1      (b) Dataset 2      (c) Dataset 3

Figure 3: GDA, Stoc-AGDA and SVRG-AGDA for robust least square.

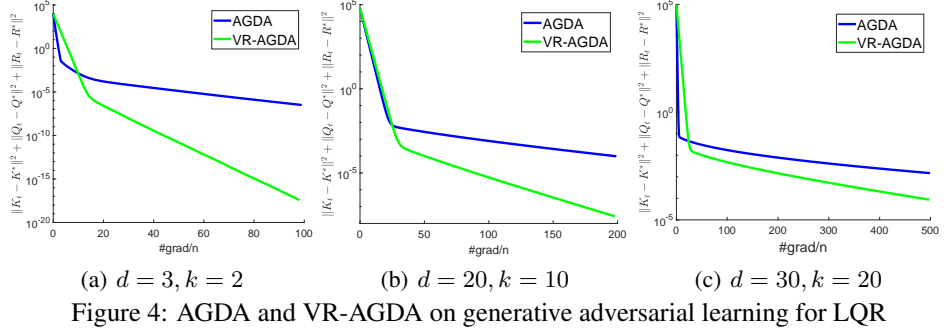

(a) $d = 3, k = 2$      (b) $d = 20, k = 10$      (c) $d = 30, k = 20$

Figure 4: AGDA and VR-AGDA on generative adversarial learning for LQR

**Remark 4.** *Theorems 4.1 and 4.2 are different in their choices of stepsizes and iteration numbers, which gives rise to different complexities. VR-AGDA with the second setting has a lower complexity than the first setting in the regime $n \leq \kappa^9$, but the first setting allows for a simpler double-loop algorithm with $T = 1$. The two theorems imply that VR-AGDA always improves over AGDA. To the best of our knowledge, this is also the first theoretical analysis of variance-reduced algorithms with alternating updating rules for minimax optimization.*

## 5 Numerical experiments

We present experiments on two applications: robust least square and imitation learning for LQR. We mainly focus on the comparison between AGDA, Stoc-AGDA, and VR-AGDA, which are the only algorithms with known theoretical guarantees. Because of their simplicity, only few hyperparameters are induced and are tuned based on grid search.

### 5.1 Robust least square

We consider the least square problems with coefficient matrix $A \in \mathbb{R}^{n \times m}$ and noisy vector $y_0 \in \mathbb{R}^n$ subject to bounded deterministic perturbation $\delta$. Robust least square (RLS) minimizes the worst case residual, and can be formulated as [16]: $\min_x \max_{\delta:\|\delta\|\leq\rho} \|Ax - y\|^2$, where $\delta = y_0 - y$. We consider RLS with soft constraint:

$$\min_x \max_y F(x,y) := \|Ax - y\|_M^2 - \lambda\|y - y_0\|_M^2, \qquad (8)$$

where we adopt the general M-(semi-)norm in: $\|x\|_M^2 = x^T M x$ and $M$ is positive semi-definite. $F(x,y)$ satisfies the two-sided PL condition when $\lambda > 1$, because it can be written as the composition of a strongly-convex-strongly-concave function and an affine function (Example 2). However, $F(x,y)$ is not strongly convex about $x$, and when $M$ is not full-rank, it is not strongly concave about $y$.

**Datasets.** We use three datasets in the experiments, and two of them are generated in the same way as in Du and Hu [15]. We generate the first dataset with $n = 1000$ and $m = 500$ by sampling rows of $A$ from a Gaussian $\mathcal{N}(0, I_n)$ distribution and setting $y_0 = Ax^* + \epsilon$ with $x^*$ from Gaussian $\mathcal{N}(0, 1)$ and $\epsilon$ from Gaussian $\mathcal{N}(0, 0.01)$. We set $M = I_n$ and $\lambda = 3$. The second dataset is the rescaled aquatic toxicity dataset by Cassotti et al. [6], which uses 8 molecular descriptors of 546 chemicals to predict quantitative acute aquatic toxicity towards Daphnia Magna. We use $M = I$ and $\lambda = 2$ for this dataset. The third dataset is generated with $A \in \mathbb{R}^{1000 \times 500}$ from Gaussian $\mathcal{N}(0, \Sigma)$ where $\Sigma_{i,j} = 2^{-|i-j|/10}$, $M$ being rank-deficit with positive eigenvalues sampled from $[0.2, 1.8]$ and $\lambda = 1.5$. These three datasets represent cases with low, median, and high condition numbers, respectively.

**Evaluation.** We compare four algorithms: AGDA, Stoc-AGDA, VR-AGDA and extragradient (EG) with fine-tuned stepsizes. For Stoc-AGDA, we choose constant stepsizes to form a fair comparison with the other two. We report the potential function value, i.e., $P_t$ described in our theorems, and distance to the limit point $\|(x_t, y_t) - (x^*, y^*)\|^2$. These errors are plotted against the number of gradient evaluations normalized by $n$ (i.e., number of full gradients). Results are reported in Figure 3. We observe that VR-AGDA and AGDA both exhibit linear convergence, and the speedup of VR-AGDA is fairly significant when the condition number is large, whereas Stoc-AGDA progresses fast at the beginning and stagnates later on. These numerical results clearly validate our theoretical findings. EG performs poorly in this example.

## 5.2 Generative adversarial imitation learning for LQR

The optimal control problem for LQR can be formulated as [18]:

$$\underset{\pi_t}{\text{minimize}} \quad \mathbb{E}_{x_0 \sim \mathcal{D}} \sum_{t=0}^{\infty} x_t^\top Q x_t + u_t^\top R u_t \quad \text{such that} \quad x_{t+1} = A x_t + B u_t, u_t = \pi_t(x_t),$$

where $x_t \in \mathbb{R}^d$ is a state, $u_t \in \mathbb{R}^k$ is a control, $\mathcal{D}$ is the distribution of initial state $x_0$, and $\pi_t$ is a policy. It is known that the optimal policy is linear: $u_t = -K^* x_t$, where $K^* \in \mathbb{R}^{k \times d}$. If we parametrize the policy in the linear form, $u_t = -K x_t$, the problem can be written as: $\min_K C(K; Q, R) :=$ $\mathbb{E}_{x_0 \sim \mathcal{D}} \left[ \sum_{t=0}^{\infty} \left( x_t^\top Q x_t + (K x_t)^\top R (K x_t) \right) \right]$ where the trajectory is induced by LQR dynamics and policy $K$. In generative adversarial imitation learning for LQR, the trajectories induced by an expert policy $K_E$ are observed and part of the goal is to learn the cost function parameters $Q$ and $R$ from the expert. This can be formulated as a minimax problem [5]:

$$\min_K \max_{(Q,R) \in \Theta} \left\{ m(K, Q, R) := C(K; Q, R) - C(K_E; Q, R) - \Phi(Q, R) \right\},$$

where $\Theta = \{(Q, R) : \alpha_Q I \preceq Q \preceq \beta_Q I, \alpha_R I \preceq R \preceq \beta_R I\}$ and $\Phi$ is a strongly-convex regularizer. We sample $n$ initial points $x_0^{(1)}, x_0^{(2)}, ..., x_0^{(n)}$ from $\mathcal{D}$ and approximate $C(K; Q, R)$ by sample average $C_n(K; Q, R) := \frac{1}{n} \sum_{i=1}^{n} \left[ \sum_{t=0}^{\infty} \left( x_t^\top Q x_t + u_t^\top R u_t \right) \right]_{x_0 = x_0^{(i)}}$. We then consider:

$$\min_K \max_{(Q,R) \in \Theta} \{ m_n(K, Q, R) := C_n(K; Q, R) - C_n(K_E; Q, R) - \Phi(Q, R) \}. \tag{9}$$

Note that $m_n$ satisfies the PL condition in terms of $K$ [18], and $m_n$ is strongly-concave in terms of $(Q, R)$, so the function satisfies the two-sided PL condition.

In our experiment, we use $\Phi(Q, R) = \lambda(\|Q - \bar{Q}\|^2 + \|R - \bar{R}\|^2)$ for some $\bar{Q}, \bar{R}$ and $\lambda = 1$. We generate a dataset with different $n$ and $k$: (1) $d = 3, k = 2$; (2) $d = 20, k = 10$; (3) $d = 30, k = 20$. The initial distribution $\mathcal{D}$ is $\mathcal{N}(0, I_d)$ and we sample $n = 100$ initial points. The exact gradients can be computed based on the compact forms established in Fazel et al. [18], Cai et al. [5]. We compare AGDA and VR-AGDA under fine-tuned stepsizes, and track their errors in terms of $\|K_t - K^*\|^2 + \|Q_t - Q^*\|_F^2 + \|R_t - R^*\|_F^2$. The result is reported in Figure 4, which again indicates that VR-AGDA significantly outperforms AGDA.

## 6 Conclusion

In this paper, we identify a subclass of nonconvex-nonconcave minimax problems, represented by the so-called two-side PL condition, for which AGDA and Stoc-AGDA can converge to *global* saddle points. We also propose the first linearly-convergent variance-reduced AGDA algorithm that is provably faster than AGDA, for this subclass of minimax problems . We hope this work can shed some light on the understanding of nonconvex-nonconcave minimax optimization: (1) different learning rates for two players are essential in GDA algorithms with alternating updates; (2) convexity-concavity is not a watershed to guarantee global convergence of GDA algorithms.

## Acknowledgments and Disclosure of Funding

This work was supported in part by ONR grant W911NF-15-1-0479, NSF CCF-1704970, and NSF CMMI-1761699.

## Broader Impact

With the boom of neural networks in every corner of machine learning, the understanding of nonconvex optimization, especially minimax optimization, becomes increasingly important. On one hand, the surge of interest in generative adversarial networks (GAN) has brought revolutionary success in many practical applications such as face synthesis , text-to-image synthesis, text generation. On the other hand, even for the simplest algorithm such as gradient descent ascent (GDA), although widely adopted by practitioners and researchers in the filed, lack theoretical understanding. It is imperative to develop a strong fundamental understanding of the success of these simple algorithms in the nonconvex regime, both to expand the usability of the methods and to accelerate future deployment in a principled and interpretable manner.

**Theory.** This paper takes an initial and substantial step towards the understanding of nonconvex-nonconcave min-max optimization problems with "hidden convexity" as well as the convergence of the simplest alternating GDA algorithm. Despite its popularity, this algorithm has not been carefully analyzed even in the convex regime. The theory developed in this work helps explain when and why GDA performs well, how to choose stepsizes, and how to improve GDA properly. These are obviously basic yet important questions that need to be addressed in order to guide future development.

**Applications.** The downstream applications include but not limited to generative adversarial networks, the actor-critic game in reinforcement learning, robust machine learning and control, and other applications in games and social economics. This work could potentially inspire more interest in broadening the applicability of GDA in practice.

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
