[Supplementary Material]

# Appendix

## A  Proofs for Section 2

We first present several key lemmas.

**Lemma A.1** (Karimi et al. [28]). *If $f(\cdot)$ is $l$-smooth and it satisfies PL with constant $\mu$, then it also satisfies error bound (EB) condition with $\mu$, i.e.*

$$\|\nabla f(x)\| \geq \mu \|x_p - x\|, \forall x,$$

*where $x_p$ is the projection of $x$ onto the optimal set, also it satisfies quadratic growth (QG) condition with $\mu$, i.e.*

$$f(x) - f^* \geq \frac{\mu}{2} \|x_p - x\|^2, \forall x.$$

*Conversely, if $f(\cdot)$ is $l$-smooth and it satisfies EB with constant $\mu$, then it satisfies PL with constant $\mu/l$.*

From the above lemma, we easily derive that $l \geq \mu$.

**Lemma A.2** (Nouiehed et al. [47]). *In the minimax problem, when $-f(x, \cdot)$ satisfies PL condition with constant $\mu_2$ for any $x$ and $f$ satisfies Assumption 1, then the function $g(x) := \max_y f(x, y)$ is $L$-smooth with $L := l + l^2/\mu_2$ and $\nabla g(x) = \nabla_x f(x, y^*(x))$ for any $y^*(x) \in \arg\max_y f(x, y)$.*

**Lemma A.3.** *In the minimax problem 1, when the objective function $f$ satisfies Assumption 1 (Lipschitz gradient) and the two-sided PL condition with constant $\mu_1$ and $\mu_2$, then function $g(x) := \max_y f(x, y)$ satisfies the PL condition with $\mu_1$.*

*Proof.* From Lemma A.2,

$$\|\nabla g(x)\|^2 = \|\nabla_x f(x, y^*(x))\|^2.$$

Since $f(\cdot, y)$ satisfies PL condition with constant $\mu_1$, we get

$$\|\nabla g(x)\|^2 \geq 2\mu_1 [f(x, y^*(x)) - \min_{x'} f(x', y^*(x))]. \tag{10}$$

Also,

$$f(x', y^*(x)) \leq \max_y f(x', y) \implies \min_{x'} f(x', y^*(x)) \leq \min_{x'} \max_y f(x', y) = g^*. \tag{11}$$

Combining equation (10) and (11), we obtain,

$$\|\nabla g(x)\|^2 \geq 2\mu_1 (g(x) - g^*).$$

$\square$

The following lemma states that stochastic gradient descent converges linearly to the neighbourhood of the optimal set under PL condition. The proof is based on [28].

**Lemma A.4.** *Consider the optimization problem $\min_x f(x) = \mathbb{E}[F(x; \xi)]$, where $f$ is $l$-smooth and satisfies PL condition with constant $\mu$. Using the stochastic gradient descent with stepsize $\tau \leq 1/l$,*

$$x_{t+1} = x_t - \tau G(x_t, \xi_t),$$

*where*

$$\mathbb{E}[G(x, \xi) - \nabla f(x)] = 0, \qquad \mathbb{E}[\|G(x, \xi) - \nabla f(x)\|^2] \leq \sigma^2,$$

*then we have*

$$\mathbb{E}[f(x_{t+1}) - f^*] \leq (1 - \mu\tau)\mathbb{E}[f(x_t) - f^*] + \frac{l\tau^2}{2}\sigma^2.$$

*Proof.* By smoothness of $f$ we have

$$f(x_{t+1}) - f^* \leq f(x_t) + \langle \nabla f(x_t), x_{t+1} - x_t \rangle + \frac{l}{2}\|x_{t+1} - x\|^2 - f^*$$

$$= f(x_t) - \tau\langle \nabla f(x_t), G(x_t, \xi_t) \rangle + \frac{l\tau^2}{2}\|G(x_t, \xi_t)\|^2 - f^*.$$

Taking expectation of both sides, we get

$$\mathbb{E}[f(x_{t+1}) - f^*] \leq \mathbb{E}[f(x_t) - f^*] - \tau\mathbb{E}[\|\nabla f(x_t)\|^2] + \frac{l\tau^2}{2}\mathbb{E}[\|G(x_t, \xi_t)\|^2]$$

$$= \mathbb{E}[f(x_t) - f^*] - \tau\mathbb{E}[\|\nabla f(x_t)\|^2] + \frac{l\tau^2}{2}\mathbb{E}[\|\nabla f(x_t)\|^2]$$

$$+ \frac{l\tau^2}{2}\mathbb{E}[\|\nabla f(x_t) - G(x_t, \xi_t)\|^2]$$

$$\leq \mathbb{E}[f(x_t) - f^*] - \frac{\tau}{2}\mathbb{E}[\|\nabla f(x_t)\|^2] + \frac{l\tau^2}{2}\sigma^2$$

$$\leq (1 - \mu\tau)\mathbb{E}[f(x_t) - f^*] + \frac{l\tau^2}{2}\sigma^2,$$

where in the equality we use $\mathbb{E}[G(x_t, \xi_t)] = \nabla f(x_t)$, in the second inequality we use $\tau \leq 1/l$, and we use PL condition in the last inequality. □

**Proof for Lemma 2.1**.

*Proof.*     • (stationary point) $\Longrightarrow$ (saddle point): From the definition of PL condition, if $(x^*, y^*)$ is a stationary point,

$$\max_y f(x^*, y) - f(x^*, y^*) \leq \frac{1}{2\mu_2}\|\nabla_y f(x^*, y^*)\|^2 = 0,$$

$$f(x^*, y^*) - \min_x f(x, y^*) \leq \frac{1}{2\mu_1}\|\nabla_x f(x^*, y^*)\|^2 = 0,$$

so $\max_y f(x^*, y) = f(x^*, y^*) = \min_x f(x, y^*)$, and therefore $f(x^*, y^*)$ is a saddle point.

• (saddle point) $\Longrightarrow$ (global minimax point): Follow from definitions.

• (global minimax point) $\Longrightarrow$ (stationary point): If $(x^*, y^*)$ is a global minimax point, then by definition,

$$y^* \in \arg\max_y f(x^*, y^*), x^* \in \arg\min_x g(x),$$

Then by first order necessary condition, we have,

$$\nabla_y f(x^*, y^*) = 0, \nabla g(x^*) = 0,$$

Further with Lemma A.2,

$$\nabla g(x^*) = \nabla_x f(x^*, y^*) = 0$$

Thus, $(x^*, y^*)$ is a stationary point.

□

**Proposition 1.** *The function*

$$f(x, y) = x^2 + 3\sin^2 x \sin^2 y - 4y^2 - 10\sin^2 y,$$

*satisfies the two-sided PL condition with* $\mu_1 = 1/16, \mu_2 = 1/14$.

*Proof.* It is not hard to derive that $\arg\min_x f(x, y) = 0, \forall y$, and $\arg\max_y f(x, y) = 0, \forall x$, i.e. $x^*(y) = y^*(x) = 0, \forall x, y$. Therefore, $(0, 0)$ is the only saddle point. Then compute the gradients:

$$\nabla_x f(x, y) = 2x + 3\sin^2(y)\sin(2x),$$

$$\nabla_y f(x, y) = -8y + 3\sin^2(x)\sin(2y) - 10\sin(2y).$$

and

$$|\nabla_x^2 f(x, y)| = |2 + 6\sin^2(y)\cos(2x)| \leq 8,$$

$$|\nabla_y^2 f(x, y)| = |-8 + 6\sin^2(x)\cos(2y) - 20\cos(2y)| \leq 28.$$

so $f(\cdot, y)$ is $L_1$-smooth with $L_1 = 8$ for any $x$ and $f(x, \cdot)$ is $L_2$-smooth with $L_2 = 28$ for any $y$. Then note that:

$$\frac{|\nabla_x f(x,y)|}{|x - x^*(y)|} = \frac{|\nabla_x f(x,y)|}{|x|} = \frac{|2x + 3\sin^2(y)\sin(2x)|}{|x|} \geq \frac{1}{2},$$

$$\frac{|\nabla_y f(x,y)|}{|y - y^*(x)|} = \frac{|\nabla_y f(x,y)|}{|y|} = \frac{|-8y + 3\sin^2(x)\sin(2y) - 10\sin(2y)|}{|y|} \geq 2.$$

So $f(\cdot, y)$ satisfies EB with $\mu_{EB1} = 1/2$, and $-f(x, \cdot)$ satisfies EB with $\mu_{EB2} = 2$. By Lemma A.1, we have $f(\cdot, y)$ satisfies PL with constant $\mu_1 = 1/16$ and $-f(x, \cdot)$ satisfies PL with constant $\mu_1 = 1/14$.

$\square$

# B  Proofs for Section 3

Before we step into proofs for Theorem 3.1, 3.2 and 3.3, we first present a contraction theorem for each iteration.

**Theorem B.1.** *Assume Assumption 1, 2, 3 hold and $f(x, y)$ satisfies the two-sided PL condition with $\mu_1$ and $\mu_2$. Define $a_t = \mathbb{E}[g(x_t) - g^*]$ and $b_t = \mathbb{E}[g(x_t) - f(x_t, y_t)]$. If we run one iteration of Algorithm 1 with $\tau_1^t = \tau_1 \leq 1/L$ (L is specified in Lemma A.2) and $\tau_2^t = \tau_2 \leq 1/l$, then*

$$a_{t+1} + \lambda b_{t+1} \leq \max\{k_1, k_2\}(a_t + \lambda b_t) + \lambda(1 - \mu_2\tau_2)\frac{L+l}{2}\tau_1^2\sigma^2 + \frac{l}{2}\lambda\tau_2^2\sigma^2 + \frac{L}{2}\tau_1^2\sigma^2,$$

*where*

$$k_1 := 1 - \mu_1\big[\tau_1 + \lambda(1 - \mu_2\tau_2)\tau_1 - \lambda(1 + \beta)(1 - \mu_2\tau_2)(2\tau_1 + l\tau_1^2)\big], \tag{12}$$

$$k_2 := 1 - \mu_2\tau_2 + \frac{l^2\tau_1}{\mu_2\lambda} + (1 - \mu_2\tau_2)\frac{l^2}{\mu_2}\tau_1 + (1 + \frac{1}{\beta})(1 - \mu_2\tau_2)\frac{l^2}{\mu_2}(2\tau_1 + l\tau_1^2), \tag{13}$$

*and $\lambda, \beta > 0$ such that $k_1 \leq 1$.*

*Proof.* Because $g$ is $L$-smooth by Lemma A.2, we have

$$g(x_{t+1}) - g^* \leq g(x_t) - g^* + \langle \nabla g(x_t), x_{t+1} - x_t \rangle + \frac{L}{2}\|x_{t+1} - x_t\|^2$$

$$= g(x_t) - g^* - \tau_1\langle \nabla g(x_t), G_x(x_t, y_t, \xi_{t1}) \rangle + \frac{L}{2}\tau_1^2\|G_x(x_t, y_t, \xi_{t1})\|^2.$$

Taking expectation of both side and use Assumption 3, we get

$$\mathbb{E}[g(x_{t+1}) - g^*] \leq \mathbb{E}[g(x_t) - g^*] - \tau_1\mathbb{E}[\langle \nabla g(x_t), \nabla_x f(x_t, y_t) \rangle] + \frac{L}{2}\tau_1^2\mathbb{E}[\|G_x(x_t, y_t, \xi_{t1})\|^2]$$

$$\leq \mathbb{E}[g(x_t) - g^*] - \tau_1\mathbb{E}[\langle \nabla g(x_t), \nabla_x f(x_t, y_t) \rangle] + \frac{L}{2}\tau_1^2\mathbb{E}[\|\nabla_x f(x_t, y_t)\|^2] + \frac{L}{2}\tau_1^2\sigma^2$$

$$\leq \mathbb{E}[g(x_t) - g^*] - \tau_1\mathbb{E}[\langle \nabla g(x_t), \nabla_x f(x_t, y_t) \rangle] + \frac{\tau_1}{2}\mathbb{E}[\|\nabla_x f(x_t, y_t)\|^2] + \frac{L}{2}\tau_1^2\sigma^2$$

$$\leq \mathbb{E}[g(x_t) - g^*] - \frac{\tau_1}{2}\mathbb{E}\|\nabla g(x_t)\|^2 + \frac{\tau_1}{2}\mathbb{E}\|\nabla_x f(x_t, y_t) - \nabla g(x_t)\|^2 + \frac{L}{2}\tau_1^2\sigma^2, \tag{14}$$

where in the second inequality we use Assumption 3, and in the third inequality we use $\tau_1 \leq 1/L$. Because $-f(x_{t+1}, y)$ is $l$-smooth and $\mu_1$-PL, by Lemma A.4, when $\tau_1 \leq 1/l$ we have

$$\mathbb{E}[g(x_{t+1}) - f(x_{t+1}, y_{t+1})] \leq (1 - \mu_2\tau_2)\mathbb{E}[g(x_{t+1}) - f(x_{t+1}, y_t)] + \frac{l}{2}\tau_2^2\sigma^2$$

$$\leq (1 - \mu_2\tau_2)\mathbb{E}[g(x_t) - f(x_t, y_t) + f(x_t, y_t) - f(x_{t+1}, y_t) + g(x_{t+1}) - g(x_t)] + \frac{l}{2}\tau_2^2\sigma^2 \tag{15}$$

Because of lipschitz continuity of the gradient, we can bound $f(x_t, y_t) - f(x_{t+1}, y_t)$ as

$$f(x_t, y_t) - f(x_{t+1}, y_t) \leq -\langle \nabla_x f(x_t, y_t), x_{t+1} - x_t \rangle + \frac{l}{2}\|x_{t+1} - x_t\|^2$$

$$\leq \tau_1 \langle \nabla_x f(x_t, y_t), G_x(x_t, y_t, \xi_{t1}) \rangle + \frac{l}{2}\tau_1^2 \|G_x(x_t, y_t, \xi_{t1})\|^2.$$

Taking expectation of both side and use Assumption 3,

$$\mathbb{E}[f(x_t, y_t) - f(x_{t+1}, y_t)] \leq (\tau_1 + \frac{l}{2}\tau_1^2)\mathbb{E}\|\nabla_x f(x_t, y_t)\|^2 + \frac{l}{2}\tau_1^2\sigma^2. \tag{16}$$

Also from (14),

$$\mathbb{E}[g(x_{t+1}) - g(x_t)] \leq -\frac{\tau_1}{2}\mathbb{E}\|\nabla g(x_t)\|^2 + \frac{\tau_1}{2}\mathbb{E}\|\nabla_x f(x_t, y_t) - \nabla g(x_t)\|^2 + \frac{L}{2}\tau_1^2\sigma^2. \tag{17}$$

Combining (15), (16) and (17),

$$\mathbb{E}[g(x_{t+1}) - f(x_{t+1}, y_{t+1})] \leq (1 - \mu_2\tau_2)\mathbb{E}[g(x_t) - f(x_t, y_t)] + (1 - \mu_2\tau_2)(\tau_1 + \frac{l}{2}\tau_1^2)\mathbb{E}\|\nabla_x f(x_t, y_t)\|^2 -$$

$$(1 - \mu_2\tau_2)\frac{\tau_1}{2}\mathbb{E}\|\nabla g(x_t)\|^2 + (1 - \mu_2\tau_2)\frac{\tau_1}{2}\mathbb{E}\|\nabla_x f(x_t, y_t) - \nabla g(x_t)\|^2 +$$

$$(1 - \mu_2\tau_2)\frac{L+l}{2}\tau_1^2\sigma^2 + \frac{l}{2}\tau_2^2\sigma^2. \tag{18}$$

Combining (14) and (18), we have for $\forall \lambda > 0$

$$a_{t+1} + \lambda b_{t+1}$$

$$\leq a_t - \left[\frac{\tau_1}{2} + \lambda(1 - \mu_2\tau_1)\frac{\tau_1}{2}\right]\mathbb{E}\|\nabla g(x_t)\|^2 + \lambda(1 - \mu_2\tau_2)b_t +$$

$$\left[\frac{\tau_1}{2} + \lambda(1 - \mu_2\tau_2)\frac{\tau_1}{2}\right]\mathbb{E}\|\nabla_x f(x_t, y_t) - \nabla g(x_t)\|^2 + \lambda(1 - \mu_2\tau_2)\left(\tau_1 + \frac{l}{2}\tau_1^2\right)\mathbb{E}\|\nabla_x f(x_t, y_t)\|^2 +$$

$$\lambda(1 - \mu_2\tau_2)\frac{L+l}{2}\tau_1^2\sigma^2 + \frac{l}{2}\lambda\tau_2^2\sigma^2 + \frac{L}{2}\tau_1^2\sigma^2$$

$$\leq a_t - \left[\frac{\tau_1}{2} + \lambda(1 - \mu_2\tau_1)\frac{\tau_1}{2} - \lambda(1 + \beta)(1 - \mu_2\tau_2)\left(\tau_1 + \frac{l}{2}\tau_1^2\right)\right]\mathbb{E}\|\nabla g(x_t)\|^2 +$$

$$\lambda(1 - \mu_2\tau_2)b_t + \left[\frac{\tau_1}{2} + \lambda(1 - \mu_2\tau_2)\frac{\tau_1}{2} + \lambda\left(1 + \frac{1}{\beta}\right)(1 - \mu_2\tau_2)\left(\tau_1 + \frac{l}{2}\tau_1^2\right)\right]\mathbb{E}\|\nabla_x f(x_t, y_t) - \nabla g(x_t)\|^2 +$$

$$\lambda(1 - \mu_2\tau_2)\frac{L+l}{2}\tau_1^2\sigma^2 + \frac{l}{2}\lambda\tau_2^2\sigma^2 + \frac{L}{2}\tau_1^2\sigma^2, \tag{19}$$

where in the second inequality we use Young's Inequality and $\beta > 0$. Now it suffices to bound $\nabla\|g(x_t)\|^2$ and $\|\nabla_x f(x_t, y_t) - \nabla g(x_t)\|^2$ by $a_t$ and $b_t$. With Lemma A.2, we have:

$$\|\nabla_x f(x_t, y_t) - \nabla g(x_t)\|^2 = \|\nabla_x f(x_t, y_t) - \nabla_x f(x_t, y^*(x_t))\|^2 \leq l^2\|y^*(x_t) - y_t\|^2, \tag{20}$$

for any $y^*(x_t) \in \arg\max_y f(x_t, y)$. Now we fix $y^*(x_t)$ to be the projection of $y_t$ on the the set $\arg\max_y f(x_t, y)$. Because $-f(\boldsymbol{x_t}, \cdot)$ satisfies PL condition with $\mu_2$, and Lemma A.1 therefore indicates it also satisfies quadratic growth condition with $\mu_2$, i.e.

$$\|y^*(x_t) - y_t\|^2 \leq \frac{2}{\mu_2}[g(x_t) - f(x_t, y_t)], \tag{21}$$

along with (20), we get

$$\|\nabla_x f(x_t, y_t) - \nabla g(x_t)\|^2 \leq \frac{2l^2}{\mu_2}[g(x_t) - f(x_t, y_t)]. \tag{22}$$

Because $g$ satisfies PL condition with $\mu_1$ by Lemma A.3,

$$\|\nabla g(x_t)\|^2 \geq 2\mu_1[g(x_t) - g^*]. \tag{23}$$

Plug (22) and (23) into (19), we can get

$$a_{t+1} + \lambda b_{t+1} \leq \left\{ 1 - \mu_1 \left[ \tau_1 + \lambda(1 - \mu_2\tau_2)\tau_1 - \lambda(1 + \beta)(1 - \mu_2\tau_2)(2\tau_1 + l\tau_1^2) \right] \right\} a_t +$$

$$\lambda \left\{ 1 - \mu_2\tau_2 + \frac{l^2\tau_1}{\mu_2\lambda} + (1 - \mu_2\tau_2)\frac{l^2}{\mu_2}\tau_1 + (1 + \frac{1}{\beta})(1 - \mu_2\tau_2)\frac{l^2}{\mu_2}(2\tau_1 + l\tau_1^2) \right\} b_t +$$

$$\lambda(1 - \mu_2\tau_2)\frac{L + l}{2}\tau_1^2\sigma^2 + \frac{l}{2}\lambda\tau_2^2\sigma^2 + \frac{L}{2}\tau_1^2\sigma^2. \tag{24}$$

$\square$

**Proof of Theorem 3.1**

*Proof.* In the setting of Theorem 1, $\tau_1^t = \tau_1$ and $\tau_2^t = \tau_2, \forall t$. By Thoerem B.1, We only need to choose $\tau_1, \tau_2, \lambda$ and $\beta$ to let $k_1, k_2 < 1$. Here we first choose $\beta = 1$ and $\lambda = 1/10$. Then

$$k_1 = 1 - \mu_1 \left[ \tau_1 + \lambda(1 - \mu_2\tau_2)\tau_1 - \lambda(1 + \beta)(1 - \mu_2\tau_2)(2\tau_1 + l\tau_1^2) \right]$$

$$\leq 1 - \mu_1 \left\{ \tau_1 - \lambda(1 - \mu_2\tau_2)\tau_1[(1 + \beta)(2 + l\tau_1) - 1] \right\} \leq 1 - \frac{1}{2}\tau_1\mu_1, \tag{25}$$

where in the last inequality we just plug in $\beta$ and $\lambda$ and use $l\tau_1 \leq 1$. Also,

$$k_2 = 1 - \mu_2\tau_2 + \frac{l^2\tau_1}{\mu_2\lambda} + (1 - \mu_2\tau_2)\frac{l^2}{\mu_2}\tau_1 + (1 + \frac{1}{\beta})(1 - \mu_2\tau_2)\frac{l^2}{\mu_2}(2\tau_1 + l\tau_1^2)$$

$$\leq 1 - \frac{l^2\tau_1}{\mu_2} \left\{ \frac{\mu_2^2\tau_2}{\tau_1 l^2} - \frac{1}{\lambda} - (1 - \mu_2\tau_2) \left[ 1 + \left( 1 + \frac{1}{\beta} \right)(2 + l\tau_1) \right] \right\}$$

$$\leq 1 - \frac{l^2\tau_1}{\mu_2}, \tag{26}$$

where in the last inequality we plug in $\beta$ and $\lambda$ and we use $\frac{\mu_2^2\tau_2}{\tau_1 l^2} \leq 18$ by our choice of $\tau_1$. Note that $\frac{1}{2}\tau_1\mu_1 < \frac{l^2\tau_1}{\mu_2}$, because $\left( \frac{1}{2}\tau_1\mu_1 \right) / \left( \frac{l^2\tau_1}{\mu_2} \right) = \frac{\mu_1\mu_2}{2l^2} < 1$. Define $P_t := a_t + \frac{1}{10}b_t$, and by Theorem B.1,

$$P_{t+1} \leq \left( 1 - \frac{1}{2}\tau_1\mu_1 \right) P_t + \frac{(1 - \mu_2\tau_2)(L + l)\tau_1^2}{20}\sigma^2 + \frac{l\tau_2^2}{20}\sigma^2 + \frac{L\tau_1^2}{2}\sigma^2.$$

With some simple computation,

$$P_t \leq (1 - \frac{1}{2}\mu_1\tau_1)^t P_0 + \frac{(1 - \mu_2\tau_2)(L + l)\tau_1^2 + l\tau_2^2 + 10L\tau_1^2}{10\mu_1\tau_1}\sigma^2.$$

We verify that $\tau_1 \leq 1/L$ by noting: $\tau_1 \leq \frac{\mu_2^2\tau_2}{18l^2} \leq \frac{\mu_2^2}{18l^3} \leq \frac{\mu_2}{2l^2}$ and $L = l + \frac{l^2}{\mu_2} \leq \frac{2l^2}{\mu_2}$. $\square$

**Proof of Theorem 3.2**

*Proof.* The first part of Theorem 3.2 is a direct corollary of Theorem 3.1 by setting $\sigma = 0$. We show the second part by noting that

$$\|x_{t+1} - x_t\|^2 = \tau_1^2 \|\nabla_x f(x_t, y_t)\|^2, \text{ and } \|y_{t+1} - y_t\|^2 = \tau_2^2 \|\nabla_y f(x_{t+1}, y_t)\|^2. \tag{27}$$

Also,

$$\|\nabla_y f(x_{t+1}, y_t)\|^2 \leq \|\nabla_y f(x_t, y_t)\|^2 + \|\nabla_y f(x_{t+1}, y_t) - \nabla_y f(x_t, y_t)\|^2$$

$$\leq \|\nabla_y f(x_t, y_t) - \nabla_y f(x_t, y^*(x_t))\|^2 + l^2\|x_{t+1} - x_t\|^2$$

$$\leq l^2 \|y_t - y^*(x_t)\|^2 + l^2\|x_{t+1} - x_t\|^2$$

$$\leq \frac{2l^2}{\mu_2}b_t + l^2\|x_{t+1} - x_t\|^2 = \frac{2l^2}{\mu_2}b_t + l^2\tau_1^2\|\nabla_x f(x_t, y_t)\|^2, \tag{28}$$

where in the second inequality $y^*(x_t)$ is the projection of $y_t$ on the the set $\arg\max_y f(x_t, y)$ and $\nabla_y f(x_t, y^*(x_t)) = 0$, in the third inequality we use lipschtiz continuity of gradient, and in the last

inequality we use quadratic growth condition. Also,

$$
\begin{aligned}
\|\nabla_x f(x_t, y_t)\|^2 &\leq \|\nabla g(x_t)\|^2 + \|\nabla_x f(x_t, y_t) - \nabla g(x_t)\|^2 \\
&= \|\nabla g(x_t) - \nabla g(x^*)\|^2 + \|\nabla_x f(x_t, y_t) - \nabla g(x_t)\|^2 \\
&\leq L^2 \|x_t - x^*\|^2 + l^2 \|y^*(x_t) - y_t\|^2 \\
&\leq \frac{2L^2}{\mu_1} a_t + \frac{2l^2}{\mu_2} b_t,
\end{aligned}
\tag{29}
$$

where in the first equality $x^*$ is the projection of $x_t$ on the set $\arg\min_x g(x)$ and $\nabla g(x^*) = 0$, in the second inequality $y^*(x_t)$ is the projection of $y_t$ on the the set $\arg\max_y f(x_t, y)$ and $\nabla g(x_t) = \nabla_x f(x_t, y_t)$, and in the last inequality we use quadratic growth condition. Therefore with (28) and (29),

$$
\begin{aligned}
\|x_t - x^*\|^2 + \|y_t - y^*\|^2 &\leq \tau_1^2 \|\nabla_x f(x_t, y_t)\|^2 + \tau_2^2 \|\nabla_y f(x_{t+1}, y_t)\|^2 \\
&\leq (1 + \tau_2^2 l^2) \tau_1^2 \|\nabla_x f(x_t, y_t)\|^2 + \frac{2l^2}{\mu_2} \tau_2^2 b_t \\
&\leq \frac{2(1 + \tau_2^2 l^2)\tau_1^2 L^2}{\mu_1} a_t + \frac{2(1 + \tau_2^2 l^2)\tau_1^2 l^2 + 2l^2 \tau_2^2}{\mu_2} b_t \\
&\leq \left[ \frac{2(1 + \tau_2^2 l^2)\tau_1^2 L^2}{\mu_1} + \frac{20(1 + \tau_2^2 l^2)\tau_1^2 l^2 + 20l^2 \tau_2^2}{\mu_2} \right] P_0 c^t,
\end{aligned}
$$

where $c = 1 - \frac{\mu_1 \mu_2^2}{36 l^3}$. Letting $\alpha_1 = \left[ \frac{2(1+\tau_2^2 l^2)\tau_1^2 L^2}{\mu_1} + \frac{20(1+\tau_2^2 l^2)\tau_1^2 l^2 + 20l^2 \tau_2^2}{\mu_2} \right] P_0$, we have

$$
\|x_{t+1} - x_t\| + \|y_{t+1} - y_t\| \leq \sqrt{2\alpha_1} c^{t/2}.
$$

For $n \geq t$,

$$
\|x_n - x_t\| + \|y_n - y_t\| \leq \sum_{i=t}^{n-1} \|x_{i+1} - x_i\| + \|y_{i+1} - y_i\| \leq \sqrt{2\alpha_1} \sum_{i=t}^{\infty} c^{i/2} \leq \frac{\sqrt{2\alpha_1} c^{t/2}}{1 - \sqrt{c}},
$$

so $\{(x_t, y_t)\}_t$ converges and by first part of this theorem the limit $(x^*, y^*)$ must be a saddle point. Thus we have

$$
\|x_t - x^*\|^2 + \|y_t - y^*\|^2 \leq \frac{2\alpha_1}{(1 - \sqrt{c})^2} c^t = \alpha c^t P_0,
$$

with $\alpha = 2 \left[ \frac{2(1+\tau_2^2 l^2)\tau_1^2 L^2}{\mu_1} + \frac{20(1+\tau_2^2 l^2)\tau_1^2 l^2 + 20l^2 \tau_2^2}{\mu_2} \right] / (1 - \sqrt{c})^2$. $\qquad\square$

**Proof of Theorem 3.3**

*Proof.* First note that since $\tau_1^t \leq \mu_2^2 / 18l^2$, $\tau_2^t = \frac{18l^2 \beta}{\mu_2^2(\gamma + t)} = \frac{18l^2 \tau_1^t}{\mu_2^2} \leq \frac{1}{l}$. Similar to the proof of Theorem 3.1, by choosing $\beta = 1$ and $\lambda = 1/10$ in the Theorem B.1, we have $\min\{k_1, k_2\} = \frac{1}{2}\mu_1 \tau_1^t$. We prove the theorem by induction. When t = 1, it is naturally satisfied by definition of $\nu$. We assume that $P_t \leq \frac{\nu}{\gamma + t}$. Then by Theorem B.1,

$$
\begin{aligned}
P_{t+1} &\leq \left( 1 - \frac{1}{2}\mu_1 \tau_1 \right) P_t + \lambda(1 - \mu_2 \tau_2^t) \frac{L+l}{2} (\tau_1^t)^2 \sigma^2 + \frac{l}{2}\lambda(\tau_2^t)^2 \sigma^2 + \frac{L}{2}(\tau_1^t)^2 \sigma^2 \\
&\leq \frac{\gamma + t - \frac{1}{2}\mu_1 \beta}{\gamma + t} \frac{\nu}{\gamma + t} + \left[ \frac{(L+l)\beta^2}{20(\gamma + t)^2} + \frac{18^2 l^5 \beta^2}{20\mu_2^4(\gamma + t)^2} + \frac{L\beta^2}{2(\gamma + t)^2} \right] \sigma^2 \\
&\leq \frac{\gamma + t - 1}{(\gamma + t)^2} \nu - \frac{\frac{1}{2}\mu_1 \beta - 1}{(\gamma + t)^2} \nu + \left[ \frac{(L+l)\beta^2}{20(\gamma + t)^2} + \frac{18^2 l^5 \beta^2}{20\mu_2^4(\gamma + t)^2} + \frac{L\beta^2}{2(\gamma + t)^2} \right] \sigma^2 \\
&\leq \frac{\nu}{\gamma + t + 1},
\end{aligned}
\tag{30}
$$

where in the second inequality we plug in $\tau_1^t$ and $\tau_2^t$, in the last inequality we use $(\gamma+t+1)(\gamma+t-1) \leq (\gamma + t)^2$ and the fact that sum of last two terms in (30) is no greater than 0 by our choice of $\nu$. $\qquad\square$

## C  Proofs for Section 4

**Proof of Theorem 4.1**

*Proof.* Because the proof is long, we break the proof into three parts for the convenience of understanding the intuition behind it.

**Part 1**.

Consider in one outer loop $k$. Define $a_{t,j} = \mathbb{E}[g(x_{t,j}) - g^*]$, $b_{t,j} = \mathbb{E}[g(x_{t,j}) - f(x_{t,j}, y_{t,j})]$, $\tilde{a}_t = \mathbb{E}[g(\tilde{x}_t) - g^*]$ and $\tilde{b}_t = \mathbb{E}[g(\tilde{x}_t) - f(\tilde{x}_t, \tilde{y}_t)]$. We omit the subscript $t$ for now. We denote the stochastic gradients as

$$G_x(x_j, y_j) = \nabla_x f_{i_j}(x_j, y_j) - \nabla_x f_{i_j}(\tilde{x}, \tilde{y}) + \nabla_x f(\tilde{x}, \tilde{y}),$$
$$G_y(x_j, y_{j+1}) = \nabla_y f_{i_j}(x_{j+1}, y_j) - \nabla_y f_{i_j}(\tilde{x}, \tilde{y}) + \nabla_y f(\tilde{x}, \tilde{y}).$$

Note that these are unbiased stochastic gradients. Similar to the proof of Theorem B.1 (replace $\sigma^2$ in (14) ), with $\tau_1 \le 1/L$, we have

$$a_{j+1} \le a_j - \frac{\tau_1}{2}\mathbb{E}\|\nabla g(x_j)\|^2 + \frac{\tau_1}{2}\mathbb{E}\|\nabla_x f(x_j, y_j) - \nabla g(x_j)\|^2 + \frac{L}{2}\tau_1^2 \mathbb{E}\|G_x(x_j, y_j) - \nabla_x f(x_j, y_j)\|^2 \tag{31}$$

By Lemma A.4, with $\tau_2 \le 1/l$,

$$b_{j+1} \le \mathbb{E}[g(x_{j+1}) - f(x_{j+1}, y_j)] - \frac{\tau_2}{2}\mathbb{E}\|\nabla_y f(x_{j+1}, y_j)\|^2 + \frac{l}{2}\tau_2^2 \mathbb{E}\|G_y(x_{j+1}, y_j) - \nabla_y f(x_{j+1}, y_j)\|^2 \tag{32}$$

Furthermore, we bound the distance to the $\tilde{x} = x_0$ as

$$\mathbb{E}\|x_{j+1} - \tilde{x}\|^2 = \mathbb{E}\|x_j - \tau_1 G_x(x_j, y_j) - \tilde{x}\|^2$$
$$= \mathbb{E}\|x_j - \tilde{x}\|^2 + 2\mathbb{E}\langle x_j - \tilde{x}, \tau_1 \nabla_x f(x_j, y_j)\rangle + \tau_1^2 \mathbb{E}\|\nabla_x f(x_j, y_j)\|^2 + \tau_1^2 \mathbb{E}\|G_x(x_j, y_j) - \nabla_x f(x_j, y_j)\|^2$$
$$\le (1 + \tau_1 \beta_1)\mathbb{E}\|x_j - \tilde{x}\|^2 + \left(\tau_1^2 + \frac{\tau_1}{\beta_1}\right)\mathbb{E}\|\nabla_x f(x_j, y_j)\|^2 + \tau_1^2 \mathbb{E}\|G_x(x_j, y_j) - \nabla_x f(x_j, y_j)\|^2, \tag{33}$$

where in the last inequality we use Young's inequality to the inner product and $\beta_1 > 0$ is a constant which we will determine later. Similarly,

$$\mathbb{E}\|y_{j+1} - \tilde{y}\|^2 \le (1 + \tau_2 \beta_2)\mathbb{E}\|y_j - \tilde{y}\|^2 + \left(\tau_2^2 + \frac{\tau_2}{\beta_2}\right)\mathbb{E}\|\nabla_y f(x_{j+1}, y_j)\|^2 + \tau_2^2 \mathbb{E}\|G_y(x_{j+1}, y_j) - \nabla_y f(x_{j+1}, y_j)\|^2, \tag{34}$$

where in the last inequality we use Young's inequality to the inner product and $\beta_2 > 0$ is a constant.
We are going to construct a potential function

$$R_j = a_j + \lambda b_j + c_j \|x_j - \tilde{x}\|^2 + d_j \|y_j - \tilde{y}\|^2, \tag{35}$$

and we will determine $\lambda$, $c_j$ and $d_j$ later. Combine (31), (32) and (34),

$$R_{j+1} \le a_j - \frac{\tau_1}{2}\mathbb{E}\|\nabla g(x_j)\|^2 + \frac{\tau_1}{2}\mathbb{E}\|\nabla_x f(x_j, y_j) - \nabla g(x_j)\|^2 + \frac{L}{2}\tau_1^2 \mathbb{E}\|G_x(x_j, y_j) - \nabla_x f(x_j, y_j)\|^2 +$$

$$\lambda \mathbb{E}[g(x_{j+1}) - f(x_{j+1}, y_j)] - \frac{\lambda \tau_2}{2}\mathbb{E}\|\nabla_y f(x_{j+1}, y_j)\|^2 +$$

$$c_{j+1}\mathbb{E}\|x_{j+1} - \tilde{x}\|^2 + \left(d_{j+1} + \frac{\lambda l}{2}\right)\tau_2^2 \mathbb{E}\|G_y(x_{j+1}, y_j) - \nabla_y f(x_{j+1}, y_j)\|^2 +$$

$$d_{j+1}(1 + \tau_2 \beta_2)\mathbb{E}\|y_j - \tilde{y}\|^2 + d_{j+1}\left(\tau_2^2 + \frac{\tau_2}{\beta_2}\right)\mathbb{E}\|\nabla_y f(x_{j+1}, y_j)\|^2 \tag{36}$$

Then we bound the variance of the stochastic gradients,

$$\mathbb{E}\|G_y(x_{j+1}, y_j) - \nabla_y f(x_{j+1}, y_j)\|^2 = \mathbb{E}\|\nabla_y f_{i_j}(x_{j+1}, y_j) - \nabla_y f_{i_j}(\tilde{x}, \tilde{y}) + \nabla_y f(\tilde{x}, \tilde{y}) - \nabla_y f(x_{j+1}, y_j)\|^2$$
$$\le \mathbb{E}\|\nabla_y f_{i_j}(x_{j+1}, y_j) - \nabla_y f_{i_j}(\tilde{x}, \tilde{y})\|^2 \le l^2 \mathbb{E}\|x_{j+1} - \tilde{x}\|^2 + l^2 \mathbb{E}\|y_j - \tilde{y}\|^2 \tag{37}$$

where in the first inequality we use $\mathbb{E}[\nabla_y f_{i_j}(x_{j+1}, y_j) - \nabla_y f_{i_j}(\tilde{x}, \tilde{y})] = \nabla_y f(x_{j+1}, y_j) - \nabla_y f(\tilde{x}, \tilde{y})$. Similarly,

$$\mathbb{E}\|G_x(x_j, y_j) - \nabla_x f(x_j, y_j)\|^2 \leq l^2 \mathbb{E}\|x_j - \tilde{x}\|^2 + l^2 \mathbb{E}\|y_j - \tilde{y}\|^2. \tag{38}$$

Plugging (37) into (36),

$$R_{j+1} \leq a_j - \frac{\tau_1}{2}\mathbb{E}\|\nabla g(x_j)\|^2 + \frac{\tau_1}{2}\mathbb{E}\|\nabla_x f(x_j, y_j) - \nabla g(x_j)\|^2 + \frac{L}{2}\tau_1^2\mathbb{E}\|G_x(x_j, y_j) - \nabla_x f(x_j, y_j)\|^2 +$$

$$\lambda\mathbb{E}[g(x_{j+1}) - f(x_{j+1}, y_j)] - \frac{\lambda\tau_2}{2}\mathbb{E}\|\nabla_y f(x_{j+1}, y_j)\|^2 +$$

$$\left[c_{j+1} + \left(d_{j+1} + \frac{\lambda l}{2}\right)l^2\tau_2^2\right]\mathbb{E}\|x_{j+1} - \tilde{x}\|^2 +$$

$$\left[d_{j+1}(1 + \tau_2\beta_2) + \left(d_{j+1} + \frac{\lambda l}{2}\right)l^2\tau_2^2\right]\mathbb{E}\|y_j - \tilde{y}\|^2 + d_{j+1}\left(\tau_2^2 + \frac{\tau_2}{\beta_2}\right)\mathbb{E}\|\nabla_y f(x_{j+1}, y_j)\|^2. \tag{39}$$

Then we plug in (33) and rearrange,

$$R_{j+1} \leq$$

$$a_j - \frac{\tau_1}{2}\mathbb{E}\|\nabla g(x_j)\|^2 + \left[c_{j+1} + \left(d_{j+1} + \frac{\lambda l}{2}\right)l^2\tau_2^2\right]\left(\tau_1^2 + \frac{\tau_1}{\beta_1}\right)\mathbb{E}\|\nabla_x f(x_j, y_j)\|^2 + \frac{\tau_1}{2}\mathbb{E}\|\nabla_x f(x_j, y_j) - \nabla g(x_j)\|^2 +$$

$$\lambda\mathbb{E}[g(x_{j+1}) - f(x_{j+1}, y_j)] - \left[\frac{\lambda\tau_2}{2} - d_{j+1}\left(\tau_2^2 + \frac{\tau_2}{\beta_2}\right)\right]\mathbb{E}\|\nabla_y f(x_{j+1}, y_j)\|^2 +$$

$$\left[c_{j+1} + \left(d_{j+1} + \frac{\lambda l}{2}\right)l^2\tau_2^2\right](1 + \tau_1\beta_1)\mathbb{E}\|x_j - \tilde{x}\|^2 + \left[d_{j+1}(1 + \tau_2\beta_2) + \left(d_{j+1} + \frac{\lambda l}{2}\right)l^2\tau_2^2\right]\mathbb{E}\|y_j - \tilde{y}\|^2 +$$

$$\left[\frac{L}{2} + c_{j+1} + \left(d_{j+1} + \frac{\lambda l}{2}\right)l^2\tau_2^2\right]\tau_1^2\mathbb{E}\|G_x(x_j, y_j) - \nabla_x f(x_j, y_j)\|^2. \tag{40}$$

Consider the second line. Using PL condition $\|\nabla_y f(x_{j+1}, y_j)\|^2 \geq 2\mu_2[g(x_{j+1}) - f(x_{j+1}, y_j)]$ and assuming $\lambda \geq d_{j+1}(\tau_2 + 1/\beta_2)$, which we will justify later by our choices of $d_{j+1}$ and $\beta_2$, we have

$$\text{the second line} \leq \lambda\left[1 - \tau_2\mu_2 + \frac{\lambda}{2}d_{j+1}\left(\tau_2^2 + \frac{\tau_2}{\beta_2}\right)\mu_2\right]\mathbb{E}[g(x_{j+1}) - f(x_{j+1}, y_j)]$$

$$\leq \lambda\left[1 - \tau_2\mu_2 + \frac{\lambda}{2}d_{j+1}\left(\tau_2^2 + \frac{\tau_2}{\beta_2}\right)\mu_2\right]\left\{b_j + \mathbb{E}(f(x_j, y_j) - f(x_{j+1}, y_j)) + (a_{j+1} - a_j)\right\}$$

$$\leq \lambda\left[1 - \tau_2\mu_2 + \frac{\lambda}{2}d_{j+1}\left(\tau_2^2 + \frac{\tau_2}{\beta_2}\right)\mu_2\right]\left\{b_j + \left(\tau_1 + \frac{l}{2}\tau_1^2\right)\mathbb{E}\|\nabla_x f(x_j, y_j)\|^2 +\right.$$

$$\frac{l}{2}\tau_1^2\mathbb{E}\|G_x(x_j, y_j) - \nabla_x f(x_j, y_j)\|^2 - \frac{\tau_1}{2}\mathbb{E}\|\nabla g(x_j)\|^2 +$$

$$\left.\frac{\tau_1}{2}\mathbb{E}\|\nabla_x f(x_j, y_j) - \nabla g(x_j)\|^2 + \frac{L}{2}\tau_1^2\mathbb{E}\|G_x(x_j, y_j) - \nabla_x f(x_j, y_j)\|^2\right\},$$

where in the last inequality we use (31) and (16). Now we plug this into $R_{j+1}$,

$$R_{j+1} \leq$$

$$a_j - \frac{\tau_1}{2}(1 + \lambda\zeta)\mathbb{E}\|\nabla g(x_j)\|^2 + \left\{\left[c_{j+1} + \left(d_{j+1} + \frac{\lambda l}{2}\right)l^2\tau_2^2\right]\left(\tau_1^2 + \frac{\tau_1}{\beta_1}\right) + \lambda\zeta\left(\tau_1 + \frac{l}{2}\tau_1^2\right)\right\}\mathbb{E}\|\nabla_x f(x_j, y_j)\|^2 +$$

$$\frac{\tau_1}{2}(1 + \lambda\zeta)\mathbb{E}\|\nabla_x f(x_j, y_j) - \nabla g(x_j)\|^2 + \lambda\zeta b_j +$$

$$\left[c_{j+1} + \left(d_{j+1} + \frac{\lambda l}{2}\right)l^2\tau_2^2\right](1 + \tau_1\beta_1)\mathbb{E}\|x_j - \tilde{x}\|^2 + \left[d_{j+1}(1 + \tau_2\beta_2) + \left(d_{j+1} + \frac{\lambda l}{2}\right)l^2\tau_2^2\right]\mathbb{E}\|y_j - \tilde{y}\|^2 +$$

$$\left[\frac{L}{2} + c_{j+1} + \left(d_{j+1} + \frac{\lambda l}{2}\right)l^2\tau_2^2 + \lambda\zeta\frac{L + l}{2}\right]\tau_1^2\mathbb{E}\|G_x(x_j, y_j) - \nabla_x f(x_j, y_j)\|^2, \tag{41}$$

where we define $\zeta = 1 - \tau_2\mu_2 + \frac{\lambda}{2}d_{j+1}\left(\tau_2^2 + \frac{\tau_2}{\beta_2}\right)\mu_2$ and $\psi = 1 - \zeta$. With $\|\nabla_x f(x_j, y_j)\|^2 \leq 2\|\nabla g(x_j)\|^2 + 2\|\nabla g(x_j) - \nabla_x f(x_j, y_j)\|^2$, we have

$R_{j+1} \leq$

$$a_j - \left\{ \frac{\tau_1}{2}(1 + \lambda\zeta) - 2 \left[ c_{j+1} + \left( d_{j+1} + \frac{\lambda l}{2} \right) l^2 \tau_2^2 \right] \left( \tau_1^2 + \frac{\tau_1}{\beta_1} \right) - 2\lambda\zeta \left( \tau_1 + \frac{l}{2}\tau_1^2 \right) \right\} \mathbb{E}\|\nabla g(x_j)\|^2 +$$

$$\lambda\zeta b_j + \left\{ \frac{\tau_1}{2}(1 + \lambda\zeta) + 2 \left[ c_{j+1} + \left( d_{j+1} + \frac{\lambda l}{2} \right) l^2 \tau_2^2 \right] \left( \tau_1^2 + \frac{\tau_1}{\beta_1} \right) - 2\lambda\zeta \left( \tau_1 + \frac{l}{2}\tau_1^2 \right) \right\} \mathbb{E}\|\nabla_x f(x_j, y_j) - \nabla g(x_j)\|^2 +$$

$$\left[ c_{j+1} + \left( d_{j+1} + \frac{\lambda l}{2} \right) l^2 \tau_2^2 \right] (1 + \tau_1\beta_1)\mathbb{E}\|x_j - \tilde{x}\|^2 + \left[ d_{j+1}(1 + \tau_2\beta_2) + \left( d_{j+1} + \frac{\lambda l}{2} \right) l^2 \tau_2^2 \right] \mathbb{E}\|y_j - \tilde{y}\|^2 +$$

$$\left[ \frac{L}{2} + c_{j+1} + \left( d_{j+1} + \frac{\lambda l}{2} \right) l^2 \tau_2^2 + \lambda\zeta \frac{L+l}{2} \right] \tau_1^2 \mathbb{E}\|G_x(x_j, y_j) - \nabla_x f(x_j, y_j)\|^2. \tag{42}$$

Then plugging in (22), (23) and (38), we get

$R_{j+1} \leq$

$$a_j - \left\{ \tau_1(1 + \lambda\zeta) - 4 \left[ c_{j+1} + \left( d_{j+1} + \frac{\lambda l}{2} \right) l^2 \tau_2^2 \right] \left( \tau_1^2 + \frac{\tau_1}{\beta_1} \right) - 4\lambda\zeta \left( \tau_1 + \frac{l}{2}\tau_1^2 \right) \right\} \mu_1 a_j +$$

$$\lambda b_j - \lambda \frac{1}{\lambda} \left\{ \lambda\psi - \frac{l^2\tau_1}{\mu_2}(1 + \lambda\zeta) - \frac{4l^2}{\mu_2} \left[ c_{j+1} + \left( d_{j+1} + \frac{\lambda l}{2} \right) l^2 \tau_2^2 \right] \left( \tau_1^2 + \frac{\tau_1}{\beta_1} \right) - \frac{4l^2}{\mu_2}\lambda\zeta \left( \tau_1 + \frac{l}{2}\tau_1^2 \right) \right\} b_j +$$

$$\left\{ \left[ c_{j+1} + \left( d_{j+1} + \frac{\lambda l}{2} \right) l^2 \tau_2^2 \right] (1 + \tau_1\beta_1) + \left[ \frac{L}{2} + c_{j+1} + \left( d_{j+1} + \frac{\lambda l}{2} \right) l^2 \tau_2^2 + \lambda\zeta \frac{L+l}{2} \right] \tau_1^2 l^2 \right\} \mathbb{E}\|x_j - \tilde{x}\|^2 +$$

$$\left\{ \left[ d_{j+1}(1 + \tau_2\beta_2) + \left( d_{j+1} + \frac{\lambda l}{2} \right) l^2 \tau_2^2 \right] + \left[ \frac{L}{2} + c_{j+1} + \left( d_{j+1} + \frac{\lambda l}{2} \right) l^2 \tau_2^2 + \lambda\zeta \frac{L+l}{2} \right] \tau_1^2 l^2 \right\} \mathbb{E}\|y_j - \tilde{y}\|^2. \tag{43}$$

Now we are ready to define sequences $\{c_j\}_j$ and $\{d_j\}_j$. Let $c_N = d_N = 0$, and

$$c_j = \left[ c_{j+1} + \left( d_{j+1} + \frac{\lambda l}{2} \right) l^2 \tau_2^2 \right] (1 + \tau_1\beta_1) + \left[ \frac{L}{2} + c_{j+1} + \left( d_{j+1} + \frac{\lambda l}{2} \right) l^2 \tau_2^2 + \lambda\zeta \frac{L+l}{2} \right] \tau_1^2 l^2,$$

$$d_j = \left[ d_{j+1}(1 + \tau_2\beta_2) + \left( d_{j+1} + \frac{\lambda l}{2} \right) l^2 \tau_2^2 \right] + \left[ \frac{L}{2} + c_{j+1} + \left( d_{j+1} + \frac{\lambda l}{2} \right) l^2 \tau_2^2 + \lambda\zeta \frac{L+l}{2} \right] \tau_1^2 l^2.$$

We further define

$$m_j^1 := \tau_1(1 + \lambda\zeta) - 4 \left[ c_{j+1} + \left( d_{j+1} + \frac{\lambda l}{2} \right) l^2 \tau_2^2 \right] \left( \tau_1^2 + \frac{\tau_1}{\beta_1} \right) - 4\lambda\zeta \left( \tau_1 + \frac{l}{2}\tau_1^2 \right), \tag{44}$$

$$m_j^2 := \frac{1}{\lambda} \left\{ \lambda\psi - \frac{l^2\tau_1}{\mu_2}(1 + \lambda\zeta) - \frac{4l^2}{\mu_2} \left[ c_{j+1} + \left( d_{j+1} + \frac{\lambda l}{2} \right) l^2 \tau_2^2 \right] \left( \tau_1^2 + \frac{\tau_1}{\beta_1} \right) - \frac{4l^2}{\mu_2}\lambda\zeta \left( \tau_1 + \frac{l}{2}\tau_1^2 \right) \right\}. \tag{45}$$

Then we can write (43) as

$$R_{j+1} \leq R_j - m_j^1 a_j - \lambda m_j^2 b_j \tag{46}$$

Now we bring back the subscript $t$. Summing the equation from 0 to $N-1$,

$$\sum_{j=0}^{N-1} a_{t,j} + \lambda b_{t,j} \leq \frac{R_0 - R_N}{N\gamma} = \frac{a_{t,0} + \lambda b_{t,0} - a_{t,N} - \lambda b_{t,N}}{N\gamma} = \frac{\tilde{a}_t + \lambda\tilde{b}_t - \tilde{a}_{t+1} - \lambda\tilde{b}_{t+1}}{N\gamma}, \tag{47}$$

where $\gamma := \min_j\{m_j^1, m_j^2\}$, and the first equality is due to $c_N = d_N = 0$ and $(x_{t,0}, y_{t,0}) = (\tilde{x}_t, \tilde{y}_t)$. Summing $t$ from 0 to $T-1$, we get

$$\frac{1}{NT} \sum_{t=0}^{T-1} \sum_{j=0}^{N-1} a_{t,j} + \lambda b_{t,j} \leq \frac{\tilde{a}_0 + \lambda\tilde{b}_0}{NT\gamma} = \frac{a^k + \lambda b^k}{NT\gamma}. \tag{48}$$

The left hand side is exactly $a^{k+1} + \lambda b^{k+1}$, because $(x_k, y_k)$ is sampled uniformly from $\{\{(x_{t,j}, y_{t,j})\}_{j=0}^{N-1}\}_{t=0}^{T-1}$.

**Part 2.**

It suffices to choose proper $\tau_1$, $\tau_2$, $N$ and $T$ such that $NT\gamma > 1$. Driven by the proof, we choose

$$\tau_1 = \frac{k_1}{\kappa^2 l}, \quad \beta_1 = k_2\kappa^2 l, \quad \tau_2 = \frac{k_3}{l}, \quad \beta_2 = lk_4.$$

We will choose $k_1, k_2, k_3$ and $k_4$ later and we let $k_1, k_2, k_3, k_4 \leq 1$. Plug back to $c_j$ and $d_j$, we have

$$c_j = \left(1 + k_1 k_2 + \frac{k_1^2}{\kappa^4}\right) c_{j+1} + \left[k_3^2(1 + k_1 k_2) + \frac{k_1^2 k_3^2}{\kappa^4} + (L+l)\frac{k_1^2}{\kappa^4}\left(\frac{k_3^2}{l^2} + \frac{k_3}{l^2 k_4}\right)\mu_2\right] d_{j+1} +$$

$$\frac{\lambda}{2} l k_3^2 (1 + k_1 k_2) + \frac{L}{2\kappa^4} k_1^2 + \frac{\lambda}{2\kappa^4} l k_1^2 k_3^2 + \frac{\lambda}{2\kappa^4}(L+l)k_1^2(1 - k_3 k_4)$$

$$\leq \left(1 + k_1 k_2 + \frac{k_1^2}{\kappa^4}\right) c_{j+1} + \left(3k_3^2 + 3\frac{1}{\kappa^3}k_1^2\right) d_{j+1} + 2\lambda l k_3^2 + (1+2\lambda)\frac{l}{\kappa^3}k_1^2, \tag{49}$$

where in the last inequality we assume $k_3^2 + \frac{k_3}{k_4} \leq 1$.

$$d_j = \frac{k_1^2}{\kappa^4} c_{j+1} + \left[1 + k_3 k_4 + k_3^2 + (L+l)\frac{k_1^2}{\kappa^4}\left(\frac{k_3^2}{l^2} + \frac{k_3}{l^2 k_4}\right)\mu_2 + \frac{1}{\kappa^4}k_1^2 k_3^2\right] d_{j+1} +$$

$$\frac{\lambda}{2} l k_3^2 + \frac{L}{2\kappa^4} k_1^2 + \frac{\lambda}{2\kappa^4} l k_1^2 k_3^2 + \frac{\lambda}{2\kappa^4}(L+l)k_1^2(1 - k_3 k_4)$$

$$\leq \frac{k_1^2}{\kappa^4} c_{j+1} + \left(1 + k_3 k_4 + 2k_3^2 + \frac{3}{\kappa^3}k_1^2\right) d_{j+1} + \lambda l k_3^2 + (1+2\lambda)\frac{l}{\kappa^3}k_1^2. \tag{50}$$

We define $e_j = \max\{c_j, d_j\}$. Then combining (49) and (50), we easily get

$$e_j \leq \left(1 + k_1 k_2 + k_3 k_4 + 3k_3^2 + \frac{4}{\kappa^3}k_1^2\right) e_{j+1} + 2\lambda l k_3^2 + (1+2\lambda)\frac{l}{\kappa^3}k_1^2.$$

As $e_N = 0$, we have

$$e_0 \leq \left[2\lambda l k_3^2 + (1+2\lambda)\frac{l}{\kappa^3}k_1^2\right] \frac{\left(1 + k_1 k_2 + k_3 k_4 + 3k_3^2 + \frac{4}{\kappa^3}k_1^2\right)^N - 1}{k_1 k_2 + k_3 k_4 + 3k_3^2 + \frac{4}{\kappa^3}k_1^2}, \tag{51}$$

and note that $e_j > e_{j+1}$ so $e_j \leq e_0, \forall j$. Then we want to lower bound $\gamma$. Rearrange (44),

$$m_j^1 = \mu_1 \left\{ \tau_1(1 + \lambda - \lambda\tau_2\mu_2) - 2\lambda l^3 \tau_2^2 \left(\tau_1^2 + \frac{\tau_1}{\beta_1}\right) - 4\lambda\left(\tau_1 + \frac{l}{2}\tau_1^2\right)(1 - \tau_2\mu_2) - \right.$$

$$\left[-2\tau_1\left(\tau_2^2 + \frac{\tau_2}{\beta_2}\right)\mu_2 + 4\left(\tau_1^2 + \frac{\tau_1}{\beta_1}\right)l^2\tau_2^2 + 8\left(\tau_1 + \frac{l}{2}\tau_1^2\right)\left(\tau_2^2 + \frac{\tau_2}{\beta_2}\right)\mu_2\right] d_{j+1} -$$

$$\left. 4\left(\tau_1^2 + \frac{\tau_1}{\beta_1}\right) c_{j+1} \right\}$$

$$\geq \frac{1}{2}\tau_1\mu_1 - \left[\frac{4}{\kappa^4}k_3^2\left(k_1^2 + \frac{k_1}{k_2}\right) + \frac{10\mu_2}{\kappa^2 l}k_1\left(k_3^2 + \frac{k_3}{k_4}\right)\right] \frac{\mu_1}{l^2} d_{j+1} - \frac{4}{\kappa^4}\left(k_1^2 + \frac{k_1}{k_2}\right)\frac{\mu_1}{l^2} c_{j+1}, \tag{52}$$

where in the inequality, we use $\lambda = 1/20$ and assume that $\frac{1}{\kappa^2}k_3^2(k_1 + \frac{1}{k_2}) \leq 10$. Rearranging (45),

$$m_j^2 = \tau_2\mu_2 - \frac{l^2\tau_1}{\mu_2}\left(\frac{1}{\lambda} + 1 - \tau_2\mu_2\right) - \frac{2l^5}{\mu_2}\left(\tau_1^2 + \frac{\tau_1}{\beta_1}\right)\tau_2^2 - \frac{4l^2}{\mu_2}\left(\tau_1 + \frac{l}{2}\tau_1^2\right)(1 - \tau_2\mu_2) -$$

$$\left[\frac{2}{\lambda}\left(\tau_2^2 + \frac{\tau_2}{\beta_2}\right)\mu_2 + \frac{2}{\lambda}l^2\tau_1\left(\tau_2^2 + \frac{\tau_2}{\beta_2}\right) + \frac{4}{\lambda}\frac{l^4}{\mu_2}\tau_2^2\left(\tau_1^2 + \frac{\tau_1}{\beta_1}\right) + \frac{8l^2}{\lambda\mu_2}\left(\tau_1 + \frac{l}{2}\tau_1^2\right)\left(\tau_2^2 + \frac{\tau_2}{\beta_2}\right)\mu_2\right] d_{j+1} -$$

$$\frac{4}{\lambda}\frac{l^2}{\mu_2}\left(\tau_1^2 + \frac{\tau_1}{\beta_1}\right) c_{j+1}$$

$$\geq \frac{l^2\tau_1}{2\min\{\mu_1, \mu_2\}} - \left[200\left(k_3^2 + \frac{k_3}{k_4}\right) + \frac{80}{\kappa^2}\left(k_1^2 + \frac{k_1}{k_2}\right)\right]\frac{\mu_2}{l^2} d_{j+1} - \frac{80}{\kappa^2}\left(k_1^2 + \frac{k_1}{k_2}\right)\frac{\mu_2}{l^2} c_{j+1}. \tag{53}$$

where in the inequality we use $\lambda = 1/20$ and assume $k_1 \leq k_3/28$ and $\frac{1}{\kappa^2} k_3^2 \left( k_1 + \frac{1}{k_2} \right) \leq 1/4$. Note that $\frac{1}{2} \tau_1 \mu_1 = \frac{\mu_1}{2\kappa^2 l} k_1$ and $\frac{l^2 \tau_1}{2 \min\{\mu_1, \mu_2\}} = \frac{l}{2\kappa^2 \min\{\mu_1, \mu_2\}} k_1$. Then we have

$$m_j^1 \geq \frac{1}{\kappa^3} \left\{ \frac{1}{2} k_1 - \left[ \frac{4}{\kappa^2} k_3^2 \left( k_1^2 + \frac{k_1}{k_2} \right) + \frac{10\mu_2}{l} k_1 \left( k_3^2 + \frac{k_3}{k_4} \right) \right] \frac{d_{j+1}}{l} - \frac{4}{\kappa^2} \left( k_1^2 + \frac{k_1}{k_2} \right) \frac{c_{j+1}}{l} \right\},$$
$$(54)$$

$$m_j^2 \geq \frac{1}{\kappa} \left\{ \frac{1}{2} k_1 - \left[ \frac{80}{\kappa^2} \left( k_1^2 + \frac{k_1}{k_2} \right) + 200 \left( k_3^2 + \frac{k_3}{k_4} \right) \right] \frac{d_{j+1}}{l} - \frac{80}{\kappa^2} \left( k_1^2 + \frac{k_1}{k_2} \right) \frac{c_{j+1}}{l} \right\}. \qquad (55)$$

Letting $k_1/k_2 = k_3/k_4$ and $k_1 = \frac{1}{28} k_3$, we have

$$\gamma \geq \frac{1}{\kappa^3} \left\{ \frac{1}{56} k_3 - 360 \left( k_3^2 + \frac{k_3}{k_4} \right) \frac{e_0}{l} \right\}, \qquad (56)$$

where we use $c_j, d_j \leq e_0, \forall j$. By plugging in $k_1 = k_3/28$ and $\lambda = 1/20$ into (51), we have

$$e_0 \leq l \frac{(1 + 2k_3 k_4 + 4k_3^2)^N - 1}{k_4/k_3 + 3}. \qquad (57)$$

Plugging this into (56), we have

$$\gamma \geq \frac{1}{\kappa^3} \left[ \frac{k_3}{56} - 360 \frac{(1 + 2k_3 k_4 + 4k_3^2)^N - 1}{k_4/k_3 + 3} \left( k_3^2 + k_3/k_4 \right) \right]. \qquad (58)$$

We choose $k_4 = k_3^{1/2}$, then

$$NT\gamma \geq \frac{1}{\kappa^3} \left[ \frac{k_3}{56} - 360 \left( (1 + 2k_3^{3/2} + 4k_3^2)^N - 1 \right) \left( \frac{k_3^2 + k_3^{1/2}}{k_3^{-1/2} + 3} \right) \right] NT. \qquad (59)$$

**Part 3**.

We choose $T = 1$, $k_3 = \beta\kappa^{-6}$ and $N = \alpha(2k_3^{3/2} + 4k_3^2)^{-1} \geq \frac{\alpha}{2} k_2^{-3/2}$, where $\alpha, \beta$ is irrelevant to $n, l, \mu_1, \mu_2$. Then since $(1 + 2k_3^{3/2} + 4k_3^2)^N \leq e^\alpha$, after plugging in $N$ and $k_3$, we have

$$NT\gamma \geq \frac{1}{\kappa^3} \left[ \frac{k_3}{56} - 360(e^\alpha - 1)(2k_3) \right] \frac{\alpha}{2} k_2^{-3/2} \geq \frac{1}{2} \left[ \frac{1}{56} - 2 \times 360(e^\alpha - 1) \right] \alpha\beta^{-1/2}. \qquad (60)$$

Therefore, for choosing $\alpha$ small enough and $\beta$ small enough, we have $NT\gamma \geq 2$. Now it remains to verify several assumptions we made in the proof. The first is $\frac{k_3}{k_4} + k_3^2 \leq 1$. Since $\frac{k_3}{k_4} + k_3^2 = k_3^{1/2} + k_3^2$, this assumption easily holds when $\beta \leq 1/4$. The second assumption we want to verify is $\frac{1}{\kappa^2} k_3^2 \left( k_1 + \frac{1}{k_2} \right) \leq 1/4$. Note that

$$\frac{1}{\kappa^2} k_3^2 \left( k_1 + \frac{1}{k_2} \right) = \frac{1}{\kappa^2} k_3^2 \left( k_1 + \frac{k_3}{k_4 k_1} \right) = \frac{1}{\kappa^2} k_3^2 \left( \frac{1}{28} k_3 + 28 k_3^{-1/2} \right).$$

So this assumption can also be easily satisfied when $\beta$ is small. The last assumption we need to verify is $\lambda \geq d_{j+1} \left( \tau_2 + \frac{1}{\beta_2} \right)$. Because $d_{j+1} \leq e_0$ and (57),

$$d_{j+1} \left( \tau_2 + \frac{1}{\beta_2} \right) \leq l \frac{(1 + 2k_3 k_4 + 4k_3^2)^N - 1}{k_4/k_3 + 3} \left( \frac{k_3}{l} + \frac{1}{k_4 l} \right)$$
$$\leq \left( (1 + 2k_3 k_4 + 4k_3^2)^N - 1 \right) \left( \frac{k_3^2 + k_3^{1/2}}{k_3^{-1/2} + 3} \right)$$
$$\leq 2(e^\alpha - 1)k_3.$$

So this assumption holds when $\alpha$ and $\beta$ are small.

$\square$

**Proof of Theorem 4.2**

*Proof.* We start from Part 3 of the proof of Theorem 4.1. We now choose $k_3 = \beta n^{-2/3}$, $N = \alpha(2k_3^{3/2} + 4k_3^2)^{-1}$, and $T = \kappa^3 n^{-1/3}$ then

$$NT\gamma \geq \frac{1}{2}\left[\frac{1}{56} - 2 \times 360(e^\alpha - 1)\right]\alpha\beta^{-1/2} \tag{61}$$

Therefore, for choosing $\alpha$ small enough and $\beta$ small enough, we have $NT\gamma \geq 2$. Note that when $\kappa^3 n^{-1/3} \leq 1$, we choose $T = 1$ and the complexity is therefore $\tilde{\mathcal{O}}(n)$. Other assumptions can be easily verified by the same way as in the proof of Theorem 4.1. □

## D  AGDA for minimax problems under one-sided PL condition

We are here to show that if $-f(x, \cdot)$ satisfies PL condition with constant $\mu$ and $f(\cdot, y)$ may be nonconvex (referred to as PL game by Nouiehed et al. [47]), AGDA as presented in Algorithm 3 can find $\epsilon$-stationary point of $g(x) := \max_y f(x, y)$ within $\mathcal{O}(\epsilon^{-2})$ iterations. Note that GDmax has complexity $\mathcal{O}(\epsilon^{-2}\log(1/\epsilon))$ on minimax problems under the one-sided PL condition [47]; SGDA has complexity $\mathcal{O}(\epsilon^{-2})$ on nonconvex-strongly-concave minimax problems [31]. Here we define condition number $\kappa = \frac{\mu}{l}$ and $L$ is still defined the same as before. The proof is based on our previous analysis and Lin et al. [31].

**Definition 3.** *$x$ is $\epsilon$-stationary point of a differential function $f$ if $\mathbb{E}\|\nabla f(x)\| \leq \epsilon$.*

---

**Algorithm 3** AGDA

---
1: Input: $(x_0, y_0)$, step sizes $\tau_1 > 0, \tau_2^t > 0$
2: **for all** $t = 0, 1, 2, ..., T - 1$ **do**
3:    $x_{t+1} \leftarrow x_t - \tau_1 \nabla f_x(x_t, y_t)$
4:    $y_{t+1} \leftarrow y_t + \tau_2 \nabla f_y(x_{t+1}, y_t)$
5: **end for**
6: choose $(x^T, y^T)$ uniformly from $\{(x_t, y_t)\}_{t=0}^T$

---

**Theorem D.1.** *Suppose Assumption 1 holds and $-f(x, \cdot)$ satisfies PL condition with constant $\mu$ for any $x$. If we run Algorithm 3 with $\tau_1 = \frac{1}{20\kappa^2 l}$ and $\tau_2 = \frac{1}{l}$, then*

$$\mathbb{E}\|\nabla g(x^T)\|^2 \leq \frac{8}{T+1}[10\kappa^2 l a_0 + \kappa^2 l b_0], \tag{62}$$

*where $a_0 = g(x_0) - g^*$ and $b_0 = g(x_0) - f(x_0, y_0)$.*

*Proof.* For convenience, we still define $b_t = g(x_t) - f(x_t, y_t)$. Since it can be easily verified that $\tau_1 \leq 1/L$, by (14) and (22), we have

$$g(x_{t+1}) \leq g(x_t) - \frac{\tau_1}{2}\|\nabla g(x_t)\|^2 + \frac{\tau_1 l^2}{\mu_2}b_t. \tag{63}$$

By (18), we have

$$b_{t+1} \leq (1 - \mu_2\tau_2)b_t + (1 - \mu_2\tau_2)\left(\tau_1 + \frac{l}{2}\tau_1^2\right)\|\nabla_x f(x_t, y_t)\|^2 - $$

$$(1 - \mu_2\tau_2)\frac{\tau_1}{2}\|\nabla g(x_t)\|^2 + (1 - \mu_2\tau_2)\frac{\tau_2}{2}\|\nabla_x f(x_t, y_t) - \nabla g(x_t)\|^2$$

$$\leq (1 - \mu_2\tau_2)b_t + \left[2(1 - \mu_2\tau_2)\left(\tau_1 + \frac{l}{2}\tau_1^2\right) - (1 - \mu_2\tau_2)\frac{\tau_2}{2}\right]\|\nabla g(x_t)\|^2 + $$

$$\left[2(1 - \mu_2\tau_2)\left(\tau_1 + \frac{l}{2}\tau_1^2\right) + (1 - \mu_2\tau_2)\frac{\tau_2}{2}\right]\|\nabla_x f(x_t, y_t) - \nabla g(x_t)\|^2$$

$$\leq (1 - \mu_2\tau_2)\left[1 + \left(5\tau_1 + 2l\tau_1^2\right)\frac{l^2}{\mu_2}\right]b_t + (1 - \mu_2\tau_2)\left[\frac{3}{2}\tau_1 + l\tau_1^2\right]\|\nabla g(x_t)\|^2, \tag{64}$$

where in the second inequality we use Young's inequality, and in third inequality we use (22). We write

$$b_{t+1} = \alpha b_t + \beta\|\nabla g(x_k)\|^2 \tag{65}$$

with
$$\alpha = (1 - \mu_2\tau_2)\left[1 + \left(5\tau_1 + 2l\tau_1^2\right)\frac{l^2}{\mu_2}\right], \qquad \beta = (1 - \mu_2\tau_2)\left[\frac{3}{2}\tau_1 + l\tau_1^2\right].$$

Then
$$b_t \le \alpha^t b_0 + \beta \sum_{k=0}^{t-1} \alpha^{t-1-k}\|\nabla g(x_k)\|^2, \quad t \ge 1.$$

Plugging into (63), we have

$$g(x_{t+1}) \le g(x_t) - \frac{\tau_1}{2}\|\nabla g(x_t)\|^2 + \frac{\tau_1 l^2}{\mu_2}\alpha^t b_0 + \frac{\tau_1 l^2 \beta}{\mu_2}\sum_{k=0}^{t-1}\alpha^{t-1-k}\|\nabla g(x_k)\|^2, \quad t \ge 1. \quad (66)$$

Telescoping and rearranging,

$$\frac{\tau_1}{2}\sum_{t=0}^{T}\|\nabla g(x_t)\|^2 - \frac{\tau_1 l^2 \beta}{\mu_2}\sum_{t=1}^{T}\sum_{k=0}^{t-1}\alpha^{t-1-k}\|\nabla g(x_k)\|^2 \le g(x_0) - g(x_{T+1}) + \frac{\tau_1 l^2}{\mu_2}b_0\sum_{t=0}^{T}\alpha^t \le a_0 + \frac{\tau_1 l^2}{\mu_2(1-\alpha)}b_0$$
$$(67)$$

Considering the left hand side of (67),

$$\sum_{t=1}^{T}\sum_{k=0}^{t-1}\alpha^{t-1-k}\|\nabla g(x_k)\|^2 = \sum_{k=0}^{T-1}\sum_{t=k+1}^{T}\alpha^{t-1-k}\|\nabla g(x_k)\|^2 \le \sum_{k=0}^{T-1}\frac{1}{1-\alpha}\|\nabla g(x_k)\|^2, \quad (68)$$

and therefore,

$$\frac{\tau_1}{2}\sum_{t=0}^{T}\|\nabla g(x_t)\|^2 - \frac{\tau_1 l^2 \beta}{\mu_2}\sum_{t=0}^{T}\sum_{k=0}^{t-1}\alpha^{t-1-k}\|\nabla g(x_k)\|^2 \ge \sum_{t=0}^{T}\left\{\frac{1}{2} - \frac{l^2\beta}{\mu_2(1-\alpha)}\right\}\tau_1\|\nabla g(x_t)\|^2.$$
$$(69)$$

We note that $\beta = (1 - \mu_2\tau_2)\left[\frac{3}{2}\tau_1 + l\tau_1^2\right] \le \frac{5}{2}\tau_1$ because $l/\tau_1 \le 1$ by our choice of $\tau_1$. Also,

$$1 - \alpha = \mu_2\tau_2 - (1 - \mu_2\tau_2)\left(5\tau_1 + 2l\tau_1^2\right)\frac{l^2}{\mu_2} \ge \mu_2\tau_2 - 7(1 - \mu_2\tau_2)\frac{\tau_1 l^2}{\mu_2} \ge \frac{1}{2\kappa}, \quad (70)$$

where in the last inequality we use $\mu_2\tau_2 = 1/\kappa$ and $(1 - \mu_2\tau_2)\frac{\tau_1 l^2}{\mu_2} = (1 - 1/\kappa)/(20\kappa) \le 1/(20\kappa)$.
Plugging into (69),

$$\frac{\tau_1}{2}\sum_{t=0}^{T}\|\nabla g(x_t)\|^2 - \frac{\tau_1 l^2 \beta}{\mu_2}\sum_{t=1}^{T}\sum_{k=0}^{t-1}\alpha^{t-1-k}\|\nabla g(x_k)\|^2 \ge \frac{\tau_1}{4}\sum_{t=0}^{T}\|\nabla g(x_t)\|^2. \quad (71)$$

Combining with (67), we have

$$\frac{1}{T+1}\sum_{t=0}^{T}\|\nabla g(x_t)\|^2 \le \frac{4}{(T+1)\tau_1}\left[a_0 + \frac{\tau_1 l^2}{\mu_2(1-\alpha)}b_0\right] \le \frac{8}{T+1}[10\kappa^2 l a_0 + \kappa^2 l b_0], \quad (72)$$

where in the inequality we use $1 - \alpha \ge 1/(2\kappa)$ again.

$\square$