[Reviews · NeurIPS 2020]

Review 1

Summary and Contributions: The paper studies the convergence of AGDA for min-max problems under PL condition. Under 2-side PL condition, the authors show AGDA has linear convergence. Furthermore, the authors present a stochastic version of AGDA, which has sublinear rate. With variance reduction technique, the rate is improved to linear.

Strengths: The major contribution of the work is to show that different versions of AGDA algorithms have linear or sublinear rate of min-max problems under PL condition, which does not need to be convex-concave. Nonconvex-nonconcave min-max problem is an important subject in ML, and there haven’t been many fundamental works on this.

Weaknesses: Though there are some merits of the paper, here are a bunch of major problems of the submission: 1. PL condition is a very strong assumption. Although it does not require convexity-concavity, it is a global condition, which roughly requires similar properties of strong convexity-concavity. I agree there are some applications of min-max problems under PL condition, as mentioned in the paper, but the applications are extremely limited, and I am not sure they are important applications to ML community. In general, nonconvex-nonconcave min-max problems won’t satisfy PL condition. To this extend, the title of the paper is a bit misleading, and it should mention PL condition explicitly. 2. It is unclear to me that AGDA is a good algorithm for min-max problem, and Figure 1 and 2 are a bit misleading. I agree that AGDA is more stable than GDA, which has been shown to be a bad algorithm for min-max problems. However, AGDA is less stable than EG. For example, when f(x,y) = xy, GDA diverges, AGDA circles, and EG converges. The objectives in Figure 1 and Figure 2 are locally strongly-convex strongly-concave, thus all three algorithms should converge linearly when the step-size is small enough. 3. The linear convergence rate shown in the paper is slow. \kappa^3 is too slow a rate for linear convergence. In most of the real problems \kappa>1000, in which case the bound becomes almost meaningless. This bound should be improvable, and the \kappa^9 bound is definitely improvable. If the authors do not agree with this, then please provide a lower-bound argument. 4. The numerical experiments are a bit toy. Robust least square is convex-concave (strongly convex-strongly concave if I understand correctly). I don’t understand the objective of LQR example, but that is a tiny problem in dimension. Is it also convex-concave? 5. The PL condition number is not known in advance, but it is required in the algorithm. This may add another layer of tuning when running the algorithm in practice. =========[after rebuttal]============ Thanks for your response, and I updated the score correspondingly. Though I see some merits of the paper, the reasons that I still did not support acceptance are: 1. The applications of two-side PL conditions are limited. The citations mentioned in the response are mostly minimization problem, not saddle-point problem. The two-side PL condition is very strong, and avoids the difficulty of nonconvex-nonconcave minimax problem, i.e., cycling. 2. The bound seems to be sub-optimal to me. One can quickly show that AGDA has O(kappa^2 log(1/eps)) rate for strongly-convex-strongly-concave problem. Though the conditions are not identical, they are similar. I invite the authors to think deeply on the possibility to improve the bound. 3. The analysis presented in the paper is hard to read, in particular the variance-reduction part. I am not sure what the readers can learn from the analysis of the submission. I suggest to improve the presentation of the analysis.

Correctness: I do not check the proof, but I do not have any reasons to doubt the correctness of the proof.

Clarity: The paper is reasonably written.

Relation to Prior Work: Yes.

Reproducibility: Yes

Additional Feedback:


Review 2

Summary and Contributions: This paper studies AGDA/Stoc-AGDA under two-sides Polyak-Łojasiewicz (PL), Moreover, this paper proposes variance reduction version of AGDA and achieves better complexity results. The motivation and the results are interesting. However, my main concerns lie in the technical part. For the two-sides Polyak-Łojasiewicz (PL), the proof procedure and results can be expected by following the proof steps in the setting of one-side Polyak-Łojasiewicz for min-max problem, i.e., [35]. For the VR-AGDA part, the variance reduction technique has been well studied in nonconvex optimization under PL setting (i.e., [r1] and so on). Hence, it appears that extending these existing techniques to the minmax problem does not seem to involve much new technical development. Based on my evaluation, I suggest to weakly reject this paper. However, I am happy to improve my score, if my questions in the weaknesses part are satisfactorily addressed in the response. [35] M. Nouiehed, M. Sanjabi, T. Huang, J. D. Lee, and M. Razaviyayn. Solving a class of nonconvex min-max games using iterative first order methods. In Advances in Neural Information Processing Systems, pages 14905–14916, 2019. [r1] Z. Li and J. Li, A Simple Proximal Stochastic Gradient Method for Nonsmooth Nonconvex Optimization. In Advances in Neural Information Processing Systems, 2018. ---------------------- post rebuttal comments The authors' response clarified their technical novelty, and I am happy to improve my rating to acceptance.

Strengths: 1. The paper studies AGDA types algorithms under two-side Polyak-Łojasiewicz. The paper shows that AGDA with properly chosen constant stepsizes converges globally to a saddle point at a linear rate of O(1-\kappa^{-3})^t. 2. The paper also explores variance reduction for AGDA, and shows that VR-AGDA achieves the complexity of O((n + n^{2/3}\kappa^3)log(1/\epsilon) ), which improves the previous results.

Weaknesses: I have the following concerns/questions: 1. Since the minimization problem with PL condition (i.e., one-side PL problem) has already been studied in the literature, what exactly is the technical difficulty to generalize the analysis under one-side PL to that under two-side PL? 2. For the VR-AGDA part, since SVRG for the minimization problem under PL has already been studied in the literature, what exactly is the technical difficulty to generalize such analysis to that for VR-AGDA under two-side PL? What is new in the technical development in this paper? How is it different from the typical minimax techniques? It will be great if the authors can point out their technical novelties to specific proof steps. I am willing to improve my score if the above questions are satisfactorily addressed.

Correctness: yes

Clarity: yes

Relation to Prior Work: yes

Reproducibility: Yes

Additional Feedback: 1. it will be much better if the authors can present the update rules of SGDA, which will clarify the difference between SGDA and AGDA. 2. The global convergence is a little bit over-claimed since usually we take PL as a local property. It is fine if the authors assume two-side PL and achieve global convergence, but at least they should remark that the two-side PL usually holds only locally. Correct me if I overlooked something.


Review 3

Summary and Contributions: This is an interesting paper which studies the convergence of alternating gradient descent ascent algorithm for two-sided PL min-max optimization problem.

Strengths: The paper is well-written and the contributions are explained clearly. The initial experiments in Figure 1 and Figure 2 describes the problem very well. To the best knowledge of the reviewer, this is the first paper analyzing the two-sided PL problems rigorously.

Weaknesses: - The reviewer would appreciate some discussion on the possibility of accelerating the proposed algorithm and whether it's optimal rate. - The paper would have been improved if more discussions is included on what practical problems potentially satisfy the two sided PL assumption (other than simple examples). ======== Edits after the rebuttal: I am lowering my score due to two reasons: 1) My concern about the practicality of two-sided PL is not fully addressed. I personally would have prefered seeing one concrete example of a two-sided PL min-max problem rather than many single PL minimziation examples. 2) The readability of the variance reduced part: This was the concern of some other reviewers and when I checked again, I agree with their comment. Also one comment that is mentioned in the discussion between the reviewers is about the specificity of the title. The title seems too generic while can be more to the point by mentioning "two-sided PL".

Correctness: The paper seem to be correct.

Clarity: Yes.

Relation to Prior Work: Some discussion on recent developments on solving nonconvex min-max problems could improve the manuscript. See the list below: + Ostrovskii, D. M., Lowy, A., & Razaviyayn, M. (2020). Efficient search of first-order nash equilibria in nonconvex-concave smooth min-max problems. arXiv preprint arXiv:2002.07919. + Zhao, R. (2020). A Primal Dual Smoothing Framework for Max-Structured Nonconvex Optimization. arXiv preprint arXiv:2003.04375. + Barazandeh, B., & Razaviyayn, M. (2020, May). Solving Non-Convex Non-Differentiable Min-Max Games Using Proximal Gradient Method. In ICASSP 2020-2020 IEEE International Conference on Acoustics, Speech and Signal Processing (ICASSP) (pp. 3162-3166). IEEE. + Lin, T., Jin, C., & Jordan, M. (2020). Near-optimal algorithms for minimax optimization. arXiv preprint arXiv:2002.02417. Chicago

Reproducibility: Yes

Additional Feedback:


Review 4

Summary and Contributions: This paper studies various alternating gradient descent ascent (AGDA) algorithms specifically for a class of nonconvex-nonconcave minimax optimization problems. The authors propose to extend the notion of Polyak-Lojasiewicz (PL) functions to bivariate functions, establishing the so-called two-sided PL condition. For objectives satisfying the two-sided PL condition, the authors proved the AGDA and its stochastic variant converge globally with a linear rate and sublinear rate respectively. A stochastic variance reduced algorithm is also proposed for problems with finite-sum structure. Numerical experiments are performed to study the convergence behavior of the proposed algorithms.

Strengths: This work studies a class of nonconvex-nonconcave minimax optimization problems, which can well be one of the earliest work in this topic, which provides global convergence guarantees and the corresponding convergence rates. The authors also propose a stochastic variance reduced AGDA algorithm which converges provably faster than AGDA.

Weaknesses: The proposal of extending the PL condition to a two-sided sense seems to be interesting, but obviously the discussion of the two-sided PL condition is not sufficient, or lacking. How more general is the two-sided PL condition than convex-concave or strongly-convex-strongly-concave? And how does it compare to one-sided PL condition + (strongly) convex/concave? The authors do not illustrate such ideas detailed enough. In addition, the choices of the numerical experiments do not reveal the necessity of introducing the two-sided PL condition as well. The authors should have chosen examples of nonconvex-nonconcave minimax optimization problems which satisfy the two-sided PL condition. Robust least squares is convex-concave, and GAIL for LQR is strongly concave in m. (Two-sided) PL condition is a more general condition which might include convex-concave functions, but for the sake of the proposal of the two-sided KL condition, examples of convex-concave, convex-nonconcave or nonconvex-concave should be excluded. Otherwise, it is a very strong deviation of the central issue this work is dealing with. Then, the authors are actually not solving nonconvex-nonconcave minimax optimization problems in the experiments. In short, the choices of the numerical experiments is not even suitable (say, to demonstrate the theoretical results).

Correctness: I did not check the proof in detail, but the claims and method seem to be correct in general. The criticism of the empirical methodology is given in the above “Weaknesses” part.

Clarity: The paper can be structured better. Especially Section 3 can be better organized. Some comments and questions: line 55: t in the big-O should be inside the parentheses line 58: nonconvex-nonconvex? line 119: “an” optimal line 129: “-”f in math mode

Relation to Prior Work: In general, the authors did a great job in explaining and discussing how this work is related to previous work and what the novel contributions of this work are.

Reproducibility: Yes

Additional Feedback: Updates after rebuttal: Thank you for the author response. I adjusted my score based on the author response, but I am still more inclined to rejection based on the following: 1. I acknowledge that the theoretical contribution of this paper is significant (faster rate of convergence than existing results) and the analysis of the AGDA algorithm seems to be novel, although I think the analysis is still hard to read (I mentioned this in my review). 2. The lack of discussion of the generality of the two-sided PL (or KL) condition is not answered in the author response. For minimization problems, the advantage of considering the KL condition is that lots of nonconvex objectives satisfy the KL condition. i.e., The KL condition is sufficiently general and we can obtain rates of convergence of first-order methods for KL functions.The authors did not demonstrate how necessary it is to consider the two-sided PL condition and did not justify whether it is the right generalization for minimax problems. In particular, I am not convinced by the argument on lines 17-20 in the author response for the motivation of considering the two-sided PL condition. The only nonconvex-nonconcave example satisfying the two-sided PL condition given in this paper, Example 1 and Figure 1, is more like a toy example which does not appear in applications to machine learning problems. 3. Following the above point and mentioned in my review, as the authors mentioned in the response, only the experiments in Figure 1 is really nonconvex-nonconcave (but they did not apply AGDA-SVRG which they propose in this paper). They also admitted in their response that LQR and robust least squares are not nonconvex-nonconcave as well. This posed a major concern of mine---even in the title of this paper, they are telling us that this paper concerns a class of nonconvex-nonconcave minimax problems (which actually satisfies the two-sided PL conditions)---but all the numerical experiments in Section 5 are not for nonconvex-nonconcave minimax problems; they are only for two-sided PL minimax problems. This gets me back to the above second point: actually are there (enough) nonconvex-nonconcave minimax problems appearing in machine learning applications which satisfy the two-sided PL conditions? I do think the authors should improve the numerical experiments really for nonconvex-nonconcave minimax problems to justify acceptance. Overall, my impression is that this paper is actually not developing results for a class of nonconvex-nonconcave minimax problems. The authors should have claimed that they are developing results for minimax problems satisfying the two-sided PL condition, instead of nonconvex-nonconcave minimax problems.

[Author Response · NeurIPS 2020]

We thank the reviewers for their constructive comments. However, we are afraid that some reviewers have underrated our technical contributions and the importance of the work.

**Technical Contributions:** While some of the ideas introduced in the paper (generalization of two-sided PL condition, algorithmic extension with variance reduction) seem intuitive, the theoretical analysis in terms of alternating GDA is nowhere near as straightforward. We emphasize several facts, which are most likely overlooked by some reviewers:

- Even in the strongly-convex-strongly-concave setting, no convergence analysis exists for alternating GDA in the literature. We provide *the first convergence result for alternating GDA* for more general problems.
- For another special case of our setting (convex-strongly-concave with bilinear term), the best known result [1] gives the $\mathcal{O}(poly(\kappa)\log(1/\epsilon))$ complexity of the simultaneous GDA, which is much worse than our result.
- Unlike simultaneous GDA, our alternating GDA uses *different learning rates* for the primal and dual variables. The ratio between these learning rates plays a critical role in establishing the convergence.
- Our analysis hinges upon novel construction of a potential function inherent from geometric property of the two-sided PL function and introducing a balancing parameter to establish the contraction. When our analysis is extended to nonconvex-PL minimax (presented in the appendix), it allows *larger stepsizes and is much neater* compared to some recent work [2], which analyzed alternating GDA in nonconvex-strongly-concave setting.

**Motivation and Importance:**

- **Why two-sided PL condition?** There are tons of applications whose minimization objectives satisfy PL condition, such as LQR (Fazel et al, 2018), computing matrix squareroot (Jain et al, 2017), phase retrieval (Zhou et al, 2016), etc. Significant advance has been recognized by utilizing the PL conditions in these field. Their robust counterparts would naturally lead to two-sided PL conditions. Hence, understanding this subclass of problems is of great importance.
- **Why alternating GDA?** Alternating GDA is widely used for training GANs and other minimax problems in practice; see e.g., [3; 4]. Yet, its convergence rate has rarely been analyzed in theory. *The main goal of this paper to is provide the theoretical analysis for alternating GDA, rather than designing an optimal algorithm for minimax optimization* (as some reviewers may expect).

**Additional Response:**

**To Reviewer 1:**  (1) See response above. (2) It may not be obvious, but the objective in Figure 1 is nonconvex-nonconcave. We will remove these figures to avoid confusion. (3) To the best our knowledge, very little is known about *algorithm-specific lower bound* for minimax optimization. Even the well-known complexity of $\mathcal{O}(\kappa^2\log(1/\epsilon))$ of simultaneous GDA on *SC-SC setting* has never been proven to be tight. (4) LQR example is non-convex in terms of the primal variable, which is a well-known challenge in the control field, albeit the small dimension.

**To Reviewer 2:**  Note that the analysis of minimization with PL condition (Karimi et al., 2016) is exactly the same as the classical analysis for strongly-convex objectives. However, there is no previous results on AGDA for SC-SC setting, and the analysis of SGDA for SC-SC setting (Facchinei and Pang, 2007) can be, by no means, extended to the two-side PL setting. Our analysis hinges upon novel construction of a potential function and leads to a number of interesting results that improve over existing work. See detailed discussion above.

**To Reviewer 3:**  Thanks for the acknowledgement of our contribution! We will dedicate a section to discuss potential accelerations of the alternating GDA algorithm and open questions related to the algorithmic-specific lower bounds.

**To Reviewer 4:**  (1) It should be noted that even in many special settings, e.g., strongly-convex-strongly-concave, convex-concave with two-sided PL, and one-sided PL+strongly-concave, our convergence analysis either improves over existing results or closes the gap in the literature. Hence, our contribution is not just limited to nonconvex-nonconcave problems with two-sided PL conditions.

(2) The example provided in Figure 1 is indeed nonconvex-nonconcave; the LQR example in Figure 4 is nonconvex-strongly-concave. Although LQR and robust least square examples include some convexity/concavity, there was no theoretical analysis before showing that AGDA achieves linear convergence for such cases.

[1] Du, S. and Hu, W. *Linear convergence of the primal-dual gradient method for convex-concave saddle point problems without strong convexity.* AISTATS, 2019.
[2] Xu, Z., Zhang, H., Xu, Y., and Lan, G. *A unified single-loop alternating gradient projection algorithm for nonconvex-concave and convex-nonconcave minimax problems.* arXiv: 2006.02032, 2020.
[3] Liu, M. and Tuzel, O. *Coupled generative adversarial networks.* NeurIPS, 2016.
[4] Metz, L., Poole, B., Pfau, D., and Sohl-Dickstein, J. *Unrolled generative adversarial networks.* ICLR, 2017.


[Meta-Review · NeurIPS 2020]

This paper studies AGDA/Stoc-AGDA for minimax problems that may not be nonconvex-nonconcave but obey the two-sides Polyak-Łojasiewicz (PL), Moreover, this paper proposes a variance reduction version of AGDA and achieves better complexity results. The reviewers thought the problem setting was interesting and relevant to Neurips but also had a variety of concerns. These concerns were partially mitigated based on the response but other concerns remained. The reviewers had a spirited and comprehensive technical discussion about the merits of this paper. Two reviewers raised their score R4 ->4-5 and R2 4->7 while one reviewer slightly lowered their score 8->7. Based on the reviews, response, discussion and my own reading the main pros and cons of this paper are as follows. I also shared this list wwith the reviewers near the end of the discussion phase and they mostly agreed with it although R1 had another perhaps more technical con which is available in their review. Pros: + Interesting theoretical contribution with faster rates of convergence for a simple algorithm + Novel analysis of the AGDA algorithm + Rigorous and interesting two-time scale analysis with a clear lower and upper-bound on the step-size that helps avoid divergence issues that may occur in the mini-max setting. Cons: - Insufficient discussion/examples on two-sided PL and when it arises (this issue was raised by all reviewers and the authors’ response did not adequately address their concerns) - The title of the paper and claims w.r.t. nonconvex-nonconcave minimax problems is a bit misleading as the central condition in the paper is two-sided PL and it is not clear how many examples/applications/numerical experiments actually obey two sided PL whilst being nonconvex-nonconcave - The exposition/proof of the variance reduction portion lacks clarity and is a bit hard to follow In my view given both the difficulty and relevance of nonconvex/nonconcave minimax problems in contemporary ML the pros outweigh the cons. Therefore, my assessment is that this paper is above the acceptance threshold.